# Variability in performance of genetic-enhanced DXA-BMD prediction models across diverse ethnic and geographic populations: A risk prediction study

**Yong Liu**[1], **Xiang-He Meng**[2], **Chong Wu**[3], **Kuan-Jui Su**[4], **Anqi Liu**[4], **Qing Tian**[4], **Lan-Juan Zhao**[4], **Chuan Qiu**[4], **Zhe Luo**[4], **Martha I Gonzalez-Ramirez**[4], **Hui Shen**[4], **Hong-Mei Xiao**[1,5]*, **Hong-Wen Deng**[4]*

**1** Center for System Biology, Data Sciences, and Reproductive Health, School of Basic Medical Science, Central South University, Changsha, Hunan Province, China, **2** Hunan Provincial Key Laboratory of Regional Hereditary Birth Defects Prevention and Control, Changsha Hospital for Maternal & Child Health Care Affiliated to Hunan Normal University, Changsha, Hunan Province, China, **3** Department of Biostatistics, The University of Texas MD Anderson Cancer Center, Houston, Texas, United States of America, **4** Tulane Center of Biomedical Informatics and Genomics, Deming Department of Medicine, School of Medicine, Tulane University, New Orleans, Louisiana, United States of America, **5** Key Laboratory of Biological, Nanotechnology of National Health Commission, Xiangya Hospital, Central South University, Changsha, Hunan Province, China

* hmxiao@csu.edu.cn (H-MX); hdeng2@tulane.edu (H-WD)

**Data Availability Statement:** The UK Biobank data utilized in this study can be accessed through an application to the UK Biobank Resource at https://www.ukbiobank.ac.uk; Data from the Osteoporotic Fractures in Men Study (MrOS, accession id: phs000373.v1.p1), the Women's Health Initiative

## Abstract

### Background

Osteoporosis is a major global health issue, weakening bones and increasing fracture risk. Dual-energy X-ray absorptiometry (DXA) is the standard for measuring bone mineral density (BMD) and diagnosing osteoporosis, but its costliness and complexity impede widespread screening adoption. Predictive modeling using genetic and clinical data offers a cost-effective alternative for assessing osteoporosis and fracture risk. This study aims to develop BMD prediction models using data from the UK Biobank (UKBB) and test their performance across different ethnic and geographical populations.

### Methods and findings

We developed BMD prediction models for the femoral neck (FNK) and lumbar spine (SPN) using both genetic variants and clinical factors (such as sex, age, height, and weight), within 17,964 British white individuals from UKBB. Models based on regression with least absolute shrinkage and selection operator (LASSO), selected based on the coefficient of determination ($R^2$) from a model selection subset of 5,973 individuals from British white population. These models were tested on 5 UKBB test sets and 12 independent cohorts of diverse ancestries, totaling over 15,000 individuals. Furthermore, we assessed the correlation of predicted BMDs with fragility fractures risk in 10 years in a case-control set of 287,183 European white participants without DXA-BMDs in the UKBB.

Clinical Trial and Observational Study (WHI, accession id: phs000200.v12.p3) and the Cardiovascular Health Study study (CHS, accession id: phs000287.v7.p1) can be requested via the database of Genotypes and Phenotypes (dbGaP) at https://www.ncbi.nlm.nih.gov/gap. The anonymized, lightweight dataset from the Louisiana Osteoporosis Study (LOS), the Kansas City Osteoporosis Study (KCOS), the China Osteoporosis Study (COS)) could obtained at Mendeley Data (https://data.mendeley.com/datasets/p78t84md5h/1). The raw whole genome array or sequencing (WGS) data will also be available on the Research Aging Biobank (https://agingresearchbiobank.nia.nih.gov).

**Funding:** H-MX was supported by the National Key Research and Development Plan of China (2017YFC1001103, 2016YFC1201805), National Natural Science Foundation of China (#81471453) (URL: https://www.nsfc.gov.cn/), and Jiangwang Educational Endowment. The funders had no role in study design, data collection and analysis, decision to publish, or preparation of the manuscript.

**Competing interests:** The authors have declared that no competing interests exist.

**Abbreviations:** BMD, bone mineral density; BMI, body mass index; CAD, coronary artery disease; CHS, Cardiovascular Health Study; CI, confidence interval; CNN, convolutional neural network; CNV, copy number variant; COS, China Osteoporosis Study; DXA, dual-energy X-ray absorptiometry; FNK, femoral neck; GD, genetic distance; GR, genetic rearrangement; GWAS, genome-wide association study; HR, hazard ratio; HRC, Haplotype Reference Consortium; KCOS, Kansas City Osteoporosis Study; LASSO, least absolute shrinkage and selection operator; LD, linkage disequilibrium; LOS, Louisiana Osteoporosis Study; LR, linear regression; MBGD, mini-batch gradient descent; MSE, mean squared error; OR, odds ratio; PC, principal component; PCA, principal component analysis; PCC, Pearson correlation coefficient; PRS, polygenic risk score; QUS, quantitative ultrasound; RA, rheumatoid arthritis; ReLU, rectified linear unit; SNP, single-nucleotide polymorphism; SPN, lumbar spine; UKBB, UK Biobank.

With single-nucleotide polymorphism (SNP) inclusion thresholds at $5\times10^{-6}$ and $5\times10^{-7}$, the prediction models for FNK-BMD and SPN-BMD achieved the highest $R^2$ of 27.70% with a 95% confidence interval (CI) of [27.56%, 27.84%] and 48.28% (95% CI [48.23%, 48.34%]), respectively. Adding genetic factors improved predictions slightly, explaining an additional 2.3% variation for FNK-BMD and 3% for SPN-BMD over clinical factors alone. Survival analysis revealed that the predicted FNK-BMD and SPN-BMD were significantly associated with fragility fracture risk in the European white population ($P < 0.001$). The hazard ratios (HRs) of the predicted FNK-BMD and SPN-BMD were 0.83 (95% CI [0.79, 0.88], corresponding to a 1.44% difference in 10-year absolute risk) and 0.72 (95% CI [0.68, 0.76], corresponding to a 1.64% difference in 10-year absolute risk), respectively, indicating that for every increase of one standard deviation in BMD, the fracture risk will decrease by 17% and 28%, respectively. However, the model's performance declined in other ethnic groups and independent cohorts. The limitations of this study include differences in clinical factors distribution and the use of only SNPs as genetic factors.

## Conclusions

In this study, we observed that combining genetic and clinical factors improves BMD prediction compared to clinical factors alone. Adjusting inclusion thresholds for genetic variants (e.g., $5\times10^{-6}$ or $5\times10^{-7}$) rather than solely considering genome-wide association study (GWAS)-significant variants can enhance the model's explanatory power. The study highlights the need for training models on diverse populations to improve predictive performance across various ethnic and geographical groups.

## Author summary

### Why was this study done?

- Osteoporosis diagnosis via bone mineral density (BMD) measurements by dual-energy X-ray absorptiometry (DXA) is impractical for large-scale screening, especially in resource-limited areas.

- Genomic data offers a cost-effective alternative for predicting disease risk, but current methods often overlook sub-significant variants and clinical factors.

- Most existing genomic prediction methods are based on European ancestry, with limited evaluation in other ethnic populations.

### What did the researchers do and find?

- We developed BMD prediction models for femoral neck (FNK) and lumbar spine (SPN) using a training set of 17,964 individuals from British white ancestry in UK Biobank (UKBB), integrating clinical and genetic factors.

- We observed that strong correlations between predicted and true BMDs ($R^2\approx25\%$ for FNK-BMD and $R^2\approx45\%$ for SPN-BMD) and significant associations with fracture risk

in European ancestry populations. And we identified the optimal $P$-value thresholds for FNK-BMD ($5\times10^{-6}$) and SPN-BMD ($5\times10^{-7}$), noting that these thresholds vary by trait and sample size.

- By applying the prediction models on 5 UKBB test sets and 12 independent cohorts of diverse ancestries, totaling over 15,000 individuals, we observed that the BMD prediction models performed well in UKBB European populations but less effectively in other ancestry groups and independent cohorts.

### What do these findings mean?

- We show that genetic factors could improve the performance of DXA-BMD prediction. Within the same population, the predicted BMDs can help prioritize individuals at high risk of fragility fracture for tailored treatments.

- Genetic prediction methods need rigorous evaluation before application to different populations, emphasizing the importance of diverse population training.

- Study limitations include differences in the distribution of clinical factors such as sex and age between the UKBB data sets and the independent cohorts, as well as the inclusion of only single-nucleotide polymorphisms (SNPs) as genetic factors.

## Introduction

Osteoporosis is a major global health problem that affects bone quality and increases the risk of fragility fractures [1]. These fractures can impair the quality of life and increase the mortality of the affected individuals, especially in the elderly population over 50 years old [2]. As the world population ages rapidly, osteoporosis and its related fractures pose a huge challenge to the health care system [3]. For example, it is estimated that the annual incidence of hip fractures will exceed 500,000 in the United States of America by 2040, and the direct medical costs for each hip fracture surgery will range from $65,000 to $68,000 [4]. Therefore, early risk prediction for osteoporosis is essential for implementing more effective intervention strategies.

The most common method for diagnosing osteoporosis and assessing the fracture risk is bone mineral density (BMD) measurement by dual-energy X-ray absorptiometry (DXA) [1]. BMD has been shown to be a reliable predictor of osteoporosis risk [1,5,6]. However, DXA has some limitations, such as high cost, low availability, and radiation exposure, which make it unsuitable for mass screening, especially in low-resource or underdeveloped regions [7]. Moreover, previous osteoporosis screening programs have revealed that only a small fraction of the screened subjects were at high risk and required early intervention, indicating that a large amount of the screening resources were spent on individuals who did not qualify for intervention [8,9]. Therefore, there is a great need for developing alternative approaches based on clinical or genetic factors to predict BMD or fracture risk, which would be beneficial for the prevention and management of osteoporosis.

BMD is determined by both genetic and environmental factors, such as sex, age, and lifestyle [10,11]. The heritability of BMD variation is estimated to range from 50% to 85%, depending on the measurement site, the age and sex of the individuals, and the population

studied [12]. With the advancement of genome detection technology, large-scale genetic studies have made remarkable progress in identifying genetic variants associated with BMD or fracture risk. In the past decade, genome-wide association studies (GWASs) have identified more than 100 loci associated with BMD, but these loci collectively explain only a small proportion (less than 20%) of the total trait heritability [11]. The variants associated with BMD are common in the population but have small effect sizes [13]. The small effect size of each variant limits its predictive value for osteoporosis. Therefore, prediction methods that combine multiple genetic variants, even with small effect sizes, could be useful for osteoporosis prediction, such as polygenic risk score (PRS) [14]. Several studies have developed and validated PRS models for BMD or fracture risk prediction using different sets of single-nucleotide polymorphisms (SNPs) and cohorts. For example, in a previous work, the researchers conducted a PRS model including 21 SNPs in 19 genes and showed that their PRS was moderately associated with nonvertebral fracture risk in Korean postmenopausal women [15]. Another study generated a PRS model based on 62 SNPs associated with femoral neck (FNK) or lumbar spine (SPN) BMD, which could only explain about 2% of the variation in BMD [16]. More recently, a study developed a PRS consisting of 21,717 genetic variants that was strongly correlated with estimated BMD of the heel (eBMD) (could explain about 23.2% of the variation in eBMD) using data from the UK Biobank (UKBB) project [17]. Despite these advancements, the existing PRS models have some limitations. First, they can only reflect a small part of the DXA-BMD variation because the current cohorts lack sufficient sample size. Second, some of them are based on eBMD, which is a surrogate measure of BMD derived from quantitative ultrasound (QUS) of the heel. Although eBMD is moderately correlated with DXA-BMD [18], it cannot reflect the BMD status of specific sites, which is crucial for assessing site-specific fracture risks [19]. Third, nongenetic risk factors are often ignored in the current PRS studies, but, for most common diseases such as osteoporosis, unglamorous but well-established risk factors like obesity, nutrition, and lifestyle may matter more than a person's genetic background [20]. Last, most of the BMD prediction models were established in European populations, and their predictive performance in other populations is still lacking evaluation [20,21]. Our study aims to address these gaps by developing prediction models that integrate genetic and clinical factors, for DXA-BMD at 2 clinically relevant sites: the femoral neck (FNK) and lumbar spine (SPN). These sites are not only pivotal for osteoporosis diagnosis but also serve as the most significant predictors of fracture risk [22]. By leveraging a larger cohort and incorporating a broader array of genetic variants, our model seeks to provide a more accurate and comprehensive reflection of BMD variation, thereby enhancing the predictive precision for osteoporosis and associated fracture risks. Further, to investigate whether the prediction models developed in the UKBB European population could be applied to other groups, we tested our models in multiple cohorts of diverse ancestry and geographic backgrounds across populations.

## Methods

### The aim and design of the study

In this study, we aimed to develop a prediction model for DXA-BMD of FNK and SPN using a large cohort from UKBB and test its performance in diverse ethnic and geographical populations. First, we used the DXA assessment data and genome-wide genetic data to build a series of prediction models to predict the DXA-BMD at FNK and SPN separately (FNK-BMD and SPN-BMD) in UKBB, which are important indicators for evaluating osteoporosis and fragility fractures risk. Then, we selected the model with best performance and assessed the predictive performance in 5 UKBB test sets and 12 independent cohorts. We also examined whether the

predicted BMDs were associated with fragility fracture risk. The workflow of our study was outlined in Fig 1.

## Study cohorts

We obtained the data sets for our study from 7 studies: the UKBB Study [23], the Louisiana Osteoporosis Study (LOS) [24], the Kansas City Osteoporosis Study (KCOS), the China Osteoporosis Study (COS), the Osteoporotic Fractures in Men Study (MrOS) [25], the Women's Health Initiative Clinical Trial and Observational Study (WHI) [26], and the Cardiovascular Health Study (CHS) [27].

The UKBB is a large-scale biomedical database and research resource containing genetic, lifestyle, and health information from more than 500,000 UK participants, enrolled at ages from 40 to 69 [28]. In total, 32,999 independent participants with DXA-BMD measurements in UKBB were used for model training and evaluation in our study. To validate the association between predicted BMDs and fragility fracture risk, we applied our prediction approaches and evaluated the association between the predicted BMDs and incidence of fragility fracture in 287,183 participants (17,490 fragility fracture cases and 269,693 controls) without DXA-BMD in UKBB. The current research project (application number 63047) was approved by UKBB.

The LOS study is an ongoing cross-sectional study with more than 17,000 subjects so far since 2011 for investigating genetic and nongenetic determinants of osteoporosis and other complex diseases/traits. In total, we included 2,863 Caucasians and 2,097 African Americans randomly selected (stratified by sex and race groups) from the whole LOS cohort [24]. We separated the cohort into LOS African Americans (LOS_AFR) set and LOS Caucasians (LOS_-CAU) set when evaluating the performance of prediction models.

The KCOS study included 2,271 unrelated subjects of European ancestry from the Kansas City osteoporosis study (KCOS_CAU). The COS study included 1,569 unrelated subjects of East Asian (Chinese Han) ancestry from the China osteoporosis study (COS_EAS). More details about the sample processing, DNA extraction as well as variant calling can be found in previous published studies [29].

The MrOS is an international multicenter longitudinal study of elderly men. The MrOS cohort recruited 5,995 men aged ≥65 years at multiple assessment centers in the USA between 2000 and 2002 [25]. Among them, we included participants with both genotype data and DXA-BMD measurements. Subsequently, we segregated them based on ancestry, creating 4 distinct data sets: 4,587 individuals of Caucasian ancestry (MrOS_CAU), 181 individuals of African American ancestry (MrOS_AFR), 165 individuals of East Asian ancestry (MrOS_EAS), and 111 individuals of Hispanic ancestry (MrOS_HIS).

The WHI study is a long-term national health investigation that has focused on strategies for preventing heart disease, breast and colorectal cancer, as well as osteoporotic fractures in postmenopausal women [26]. In this study, we included 1,064 individuals with both genotype data and DXA-BMD measurements. Subsequently, we segregated them by ancestry: 671 individuals of black or African-American ancestry (WHI_AFR) and 393 individuals of Hispanic/Latino ancestry (WHI_HIS).

The CHS is a prospective investigation aimed at identifying risk factors for the development and progression of coronary heart disease and stroke in individuals aged 65 years and older [27]. We categorized the CHS cohort by ancestry into 2 sets: 432 individuals of white/Caucasian ancestry (CHS_CAU) and 190 individuals of black/African-American ancestry (CHS_AFR).

All samples were approved by the respective institutional ethics review boards, and all participants provided written informed consent.

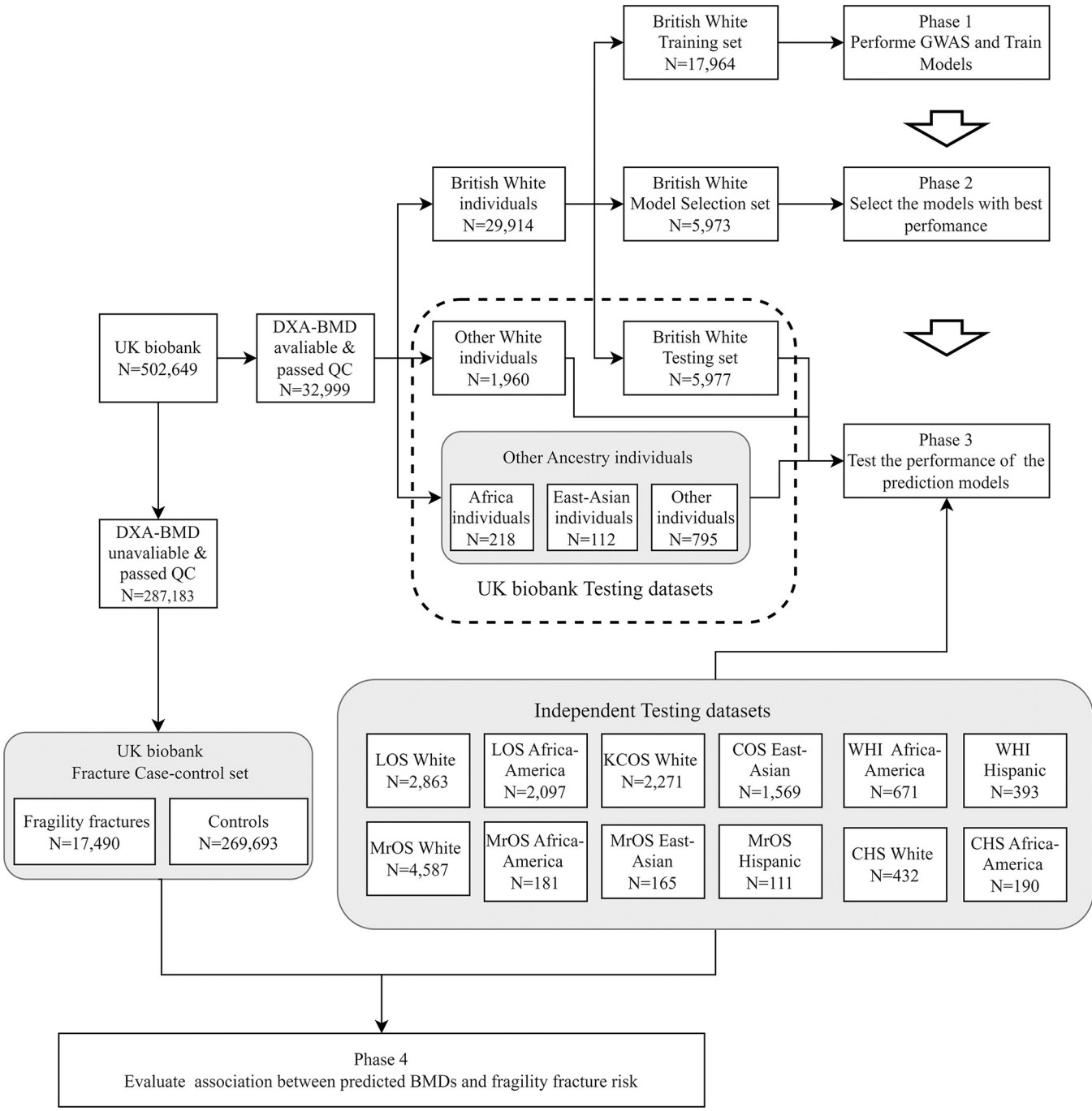

**Fig 1. The workflow of our study.** Study overview. At Phase 1, we built a series of prediction models to predict the DXA-BMD at FNK and SPN separately (FNK-BMD and SPN-BMD) in UKBB Training set. At Phase 2, we selected the model with highest $R^2$ in UKBB Model selection set as the prediction for further analysis. At Phase 3, we evaluated the predictive performance in 3 UKBB testing data sets and 12 independent data sets. At Phase 4, we examined the association between predicted BMDs and fragility fracture risk in UKBB Fragility facture case-control set and the independent data sets. BMD, bone mineral density; DXA, dual-energy X-ray absorptiometry; FNK, femoral neck; SPN, lumbar spine; UKBB, UK Biobank.

## Data preprocessing

For UKBB cohort, we extracted the participants who underwent the DXA measurement to develop the BMD prediction models. Considering the requirement for a substantial number of

samples for the model training and promotion, we have adopted a relatively stringent criterion for the exclusion of outlier samples. Specifically, we exclude samples with extreme BMD values below or above the 0.1th percentile to mitigate potential errors that may arise from handing factors. To ensure the independence of our samples, the kinship between each pair of individuals was inferred using KING software [30]. We randomly retained only 1 individual from the inferred kinship relationships within third-degree or closer, while the others were excluded. For more detailed data preprocessing procedures, please refer to S1 Appendix.

## Splitting of UKBB data

To reduce the impact of population heterogeneity, we limited the model training and selection to British white individuals, but we also tested the best-performing models in other ethnic populations. Participants of British white ancestry in UKBB, with DXA-BMD measurements, and genotyping information ($N$ = 29,914) were randomly assigned to the UKBB British Training set ($N$ = 17,964, 60% of British white individuals), the UKBB British Model Selection set ($N$ = 5,973, 20% of British white individuals), or the UKBB British Testing set ($N$ = 5,977, 20% of British white individuals). The rest of the participants with DXA-BMD measurements in UKBB were divided to UKBB other white set ($N$ = 1,960), UKBB African ancestry set ($N$ = 218), UKBB East Asian ancestry set ($N$ = 112), and UKBB other ancestry set ($N$ = 795). In addition, we included 287,183 white individuals (17,490 fragility fracture cases and 269,693 controls) without DXA-BMDs to assess the association between predicted BMDs and fragility fractures (the UKBB Fracture case-control set). The UKBB Fracture case-control set had no overlap with the UKBB British Training, Model Selection, Testing, African, East Asian, or other ancestry sets.

## Phenotype measurements and quality control

The BMD measurements were performed at an imaging assessment center for UKBB with a GE-Lunar iDXA instrument. The clinical risk factors were collected at the imaging visit (prior to DXA scan), including age, sex, height, and weight. The lifestyle variables were collected from the touchscreen questionnaire completed at the UKBB assessment center, including smoking, drinking, and exercise. The individuals in the UKBB Fracture case-control set did not attend the DXA measurement, so their variables were obtained at the initial assessment visit. Fragility fracture cases were identified based on the 10th revision of the International Statistical Classification of Diseases and Related Health Problems (ICD10) codes of primary or secondary diagnoses and self-reported codes following the initial assessment. Individuals without any hospital inpatient data were excluded from this study. The rheumatoid arthritis (RA) cases were identified based on ICD10 codes of primary or secondary diagnoses, and the glucocorticoid using information was obtained from the record-level primary care linked data. Individuals lacking confirmed sex or age information were excluded. For other missing clinical factors, we imputed the values using the median of the respective cohort. A full list of ICD10 and self-reported codes used can be found in S1 Table. More details of phenotype measurements are available in S1 Appendix.

## Processing of genetic data

Except for LOS, the genetic data from UKBB, KCOS, COS, MrOS, WHI, and CHS that were genotyped using chip technology was first imputed with the Haplotype Reference Consortium (HRC) [31], UK10K [32], or 1000 Genomes (1000G) haplotype reference panels [33]. For UKBB, the imputed genetic data was obtained from the official release, which was with the HRC and UK10K haplotype resource [31,32]. For KCOS, COS, and MrOS, we performed the

genetic imputation using the Michigan Imputation Server with the HRC and 1000G panels [32–34]. Since the genetic data in LOS was genotyped by the next-generation whole genome sequencing with an enough read depth of 22×, we did not perform imputation for this cohort. All genetic data was converted to GRCh37 version. After imputation, we applied quality control to the million imputed SNPs with the following criterions: (1) Imputation scores >0.3; (2) Minimal allele frequency >0.001; (3) Missing rate <0.1; (4) P-value of Hardy–Weinberg equilibrium test $> 1×10^{-6}$.

## Individual-level genetic distance (GD) from training data set

In the Model Selection set and Testing sets (both UKBB and independent data sets), we calculated the GD for each individual. As described in the previous literature [35], the GD is defined as the Euclidean distance between a target individual and the center of training data on the principal component (PC) space of training data. Specifically, we first performed linkage disequilibrium (LD) pruning with plink1.9 (—indep-pairwise 1000 50 0.05) (https://www.cog-genomics.org/plink) and excluded the long-range LD regions. Next, we conducted principal component analysis (PCA) with flashpca2 [36] on the UKBB British Training set to obtain the top 20 PCs. Then, we projected the individuals in the Model Selection set and Testing sets onto the PC space of training data by using SNP loadings (with their means and standard deviations) output from flashpca2. In the end, we computed the GD for each individual as the Euclidean distance of their PCs from the center of training data with the equation

$d_i = \sqrt{\sum_{j=1}^{20} (pc_{ij})^2}$, in which $pc_{ij}$ represents the $j^{th}$ PC of individual $i$.

## Genome-wide association study (GWAS)

We performed a GWAS for FNK-BMD and SPN-BMD in the UKBB Training set ($N = 17,964$). After imputation and QC, ~19 million SNPs were included in the GWAS. We used the BOLT-LMM software [37] to perform a linear mixed model to perform the association test for FNK-BMD and SPN-BMD separately, with adjustment for age, square of age, sex, height, weight, the first 20th genetic PCs, assessment center and genotyping array. Although we have utilized the largest available single ancestry DXA-BMD population to train our model, it still only encompasses a magnitude of tens of thousands, which is relatively small compared to the vast number of genomic variations. Additionally, there is a high correlation between linked SNPs. Including all of them in the model could lead to multicollinearity issues, potentially making the model difficult to train and generalize. To reduce the input features of prediction models, we obtained LD-independent associations using PLINK1.9 by clumping SNPs in LD at a squared correlation ($r^2$) > 0.05 and selected the most significant SNP from within each LD window.

## Prediction models of BMDs

In the UKBB Training set ($N = 17,964$), we built 4 prediction models that integrated with clinical and genetic factors to predict FNK-BMD and SPN-BMD, separately. These models were:

1. Clumping and Thresholding (C+T) based PRS: Initially, we derived the C+T based PRSs from the GWAS for FNK-BMD and SPN-BMD in the UKBB British Training set using PRSice-2 [38,39]. Subsequently, we integrated these calculated PRS with clinical factors to train a regression model for predicting BMDs.

2. Linear regression (LR): Differing from the C+T based PRS approach, this method does not require predetermined SNP weights from GWAS but learns them directly from the training data [40].

3. Regression with least absolute shrinkage and selection operator (LASSO) [41]: In contrast to a simple linear regression model, LASSO models incorporate a regularization parameter ($\lambda$) for both variable selection and regularization [41].

4. Convolutional neural network (CNN) [42]: The CNN model, a deep learning architecture widely used dimensionality reduction and feature recognition in image processing tasks, was employed in our study. The network architecture diagram is shown in S1 Fig. Compared to traditional regression models, neural network models with activation functions [the rectified linear unit (ReLU) in this study] are advantageous for capturing nonlinear relationships between SNPs and BMDs [43].

For each type of model, we fitted 5 models that integrated clinical factors and SNPs with $p$-values smaller than a chosen set of thresholds ($5\times10^{-8}$, $5\times10^{-7}$, $5\times10^{-6}$, $5\times10^{-5}$, and $5\times10^{-4}$). More details about the models and the parameter selection process are available in S1 Appendix, S2 and S3 Figs. For each type of model and $p$-value threshold, the model with the highest $R^2$ in UKBB British Model Selection set was then taken forward for testing in the testing sets. In total, we included 18 clinical factors in the prediction models: age, square of age, sex, height, weight, smoking, drinking, exercise, and the first 10th genetic PCs. The clinical risk factors chosen for BMD and fracture risk were determined associated with osteoporosis and osteoporotic fracture [10]. They are also the most commonly used variables in genetic studies on osteoporosis, such as sex, age, body weight, etc. Furthermore, we constructed an LR model that includes only clinical factors to evaluate whether incorporating genetic factors could improve the prediction for BMDs. The diagram that shows the structure of models is presented in S4 Fig. All continuous variables (age, square of age, height, weight, and genetic PCs) were normalized with zero-mean normalization, and categorical variables (sex, smoking, drinking, exercise, and SNPs) were encoded with one-hot encoding. The S2 Table provides detailed information on the input features of the models.

## Model training and evaluation

The model training and evaluation was performed separately for FNK-BMD and SPN-BMD. For each type of model, we trained the model in the UKBB British Training set using a series of parameters (see the Prediction models of BMDs section). All weights in the models were initialized using random initialization, and the model parameters were adjusted based on the mean squared error (MSE) [44]. Due to the large total number of variables and sample size, it is not feasible to process all data at once. Therefore, we have adopted the mini-batch gradient descent (MBGD) approach [45], with a batch size set to 32 and an initial learning rate of $10^{-4}$, utilizing the Adam optimizer [46] to adjust the learning rates of the parameters. We optimized the models by minimizing the MSE and evaluated the models with the coefficient of determination ($R^2$) and Pearson correlation coefficient (PCC) between the predicted and actual BMD values. The model conduction and evaluation were performed using Python3.9 and TensorFlow2.

## Uncertainty and sensitivity analyses

To estimate the bias and confidence intervals (CIs) of model performances, we implemented a bootstrap strategy, randomly resampling the training data set 50 times for independent training and testing. The bootstrap method is particularly robust, providing estimates of the model's mean performance and variability. Its key advantage lies in its non-reliance on rigid parametric assumptions, offering flexibility in addressing a range of statistical issues, such as nonlinear regression, determination of CIs, and evaluation of bias [47].

### Evaluating the association between predicted BMDs and fragility fracture risk

To examine the association between the predicted BMDs and fragility fracture risk, we first compared the mean value of the predicted BMDs the fragility fracture cases and controls using a *T* test in the UKBB Fracture case-control set, in which the DXA-BMD measurements were not available. Next, we grouped all individuals by different quantile ranges of predicted BMDs: ≤5%, 5%–20%, 20%–40%, 40%–60%, 60%–80%, 80%–95%, and >95%, and quantified the incidence of fragility fractures in each group.

To investigate whether the predicted BMDs were useful in predicting future fracture risk, we performed survival analyses to evaluate the cumulative incidence of fragility fracture in the UKBB Fracture case-control set, censored by 10 years. First, we performed a univariate survival analysis with the Kaplan–Meier model to examine whether there was a difference in cumulative fracture risk among different groups based on different quantile ranges of predicted BMDs. Then, we performed a multivariate survival analysis with the Cox proportional hazards regression (Cox) model, integrating the predicted BMDs and clinical factors: age, sex, height, weight, body mass index (BMI), previous fracture, smoking, drinking, exercise, glucocorticoid using, and RA. We incorporated additional variables such as glucocorticoid use and RA status into the fracture prediction model, as these factors have been proven to be closely associated with osteoporotic fractures [48,49] and are utilized in the commonly used fracture prediction tool, Fracture Risk Assessment Tool (FRAX, https://frax.shef.ac.uk/FRAX/) [50]. The hazard ratios (HRs) and 95% CIs for fracture incidence based on predicted BMDs were estimated using a Cox model, and the absolute risk was also calculated [51]. We assessed the predictive performance of the Cox model using the Concordance Index (C-index). Survival analyses were performed using the R software (version 4.1.1) with the "survival" package (version 3.2) and "survminer" package (version 4.1.2).

## Results

### Characteristics of the study population

In total, we included 320,182 individuals from UKBB for model construction and evaluation in our study, which were split to 8 data sets: the UKBB British Training set (*N* = 17,964), the UKBB British Model Selection set (*N* = 5,973), the UKBB British Test set (*N* = 5,977), the UKBB other White Test set (*N* = 1,960), the UKBB African ancestry set (*N* = 218), the UKBB East Asian ancestry set (*N* = 112), the UKBB other ancestry set (*N* = 795), and the UKBB Fracture case-control set (*N* = 287,183). We also included 12 data sets from 6 independent cohorts to evaluate the selected prediction models: LOS_CAU (*N* = 2,863), LOS_AFR (*N* = 2,097), KCOS_CAU (*N* = 2,271), COS_EAS (*N* = 1,569), MrOS_CAU (*N* = 4,587), MrOS_AFR (*N* = 181), MrOS_EAS (*N* = 165), MrOS_HIS (*N* = 111), WHI_AFR (*N* = 671), WHI_HIS (*N* = 393), CHS_CAU (*N* = 432), and CHS_AFR (*N* = 190). The demographic characteristics of each data set are summarized in Table 1. The first 2 genetic PCs of all individuals and the GD to the training data are shown in

**Table 1. Participant characteristics by data set.**

| Datasets | Sample size | Age Mean (SD) | Women N (%) | Height Mean (SD) | Weight Mean (SD) | Smoking N (%) | Drinking N (%) | Exercise N (%) | FNK-BMD mean (SD) | SPN-BMD mean (SD) | Fracture N (%) |
|---|---|---|---|---|---|---|---|---|---|---|---|
| UKBB Training | 17,964 | 63.95 (7.56) | 9,113 (50.73) | 170.56 (9.42) | 75.62 (14.99) | 6,631 (36.91) | 17,456 (97.17) | 16,289 (90.68) | 0.94 (0.14) | 1.1 (0.18) | 616 (3.43) |
| UKBB Model Selection | 5,973 | 64.15 (7.55) | 2,973 (49.77) | 170.56 (9.39) | 75.82 (15.47) | 2,273 (38.05) | 5,797 (97.05) | 5,394 (90.31) | 0.94 (0.14) | 1.1 (0.18) | 207 (3.47) |
| UKBB British Test | 5,977 | 63.82 (7.6) | 3,082 (51.56) | 170.47 (9.3) | 75.68 (15.18) | 2,286 (38.25) | 5,808 (97.17) | 5,405 (90.43) | 0.94 (0.14) | 1.1 (0.18) | 212 (3.55) |

*(Continued)*

**Table 1.** (Continued)

| Datasets | Sample size | Age Mean (SD) | Women N (%) | Height Mean (SD) | Weight Mean (SD) | Smoking N (%) | Drinking N (%) | Exercise N (%) | FNK-BMD mean (SD) | SPN-BMD mean (SD) | Fracture N (%) |
|---|---|---|---|---|---|---|---|---|---|---|---|
| UKBB Other White | 1,960 | 62.86 (7.75) | 1,086 (55.41) | 169.88 (9.48) | 75.13 (15.81) | 890 (45.41) | 1,886 (96.22) | 1,785 (91.07) | 0.92 (0.14) | 1.08 (0.18) | 75 (3.83) |
| UKBB AFR | 218 | 58.41 (7.09) | 117 (53.67) | 169.5 (9.36) | 80.33 (17.33) | 68 (31.19) | 191 (87.61) | 186 (85.32) | 1.05 (0.16) | 1.21 (0.19) | 3 (1.38) |
| UKBB EAS | 112 | 59.96 (6.99) | 63 (56.25) | 163.82 (7.91) | 62.5 (11.2) | 23 (20.54) | 96 (85.71) | 91 (81.25) | 0.9 (0.15) | 1.06 (0.19) | 2 (1.79) |
| UKBB Other Ancestry | 795 | 61.05 (8.18) | 390 (49.06) | 166.93 (9.47) | 72.43 (14.33) | 246 (30.94) | 650 (81.76) | 689 (86.67) | 0.95 (0.14) | 1.1 (0.18) | 13 (1.64) |
| UKBB Case-control | 287,183 | 57.1 (8.02) | 154,601 (53.83) | 168.75 (9.2) | 78.68 (16.02) | 136,125 (47.4) | 277,919 (96.77) | 175,539 (61.12) | NA | NA | 17,490 (6.09) |
| LOS_CAU | 2,863 | 38.83 (12.3) | 1,510 (52.74) | 169.59 (9.23) | 76.25 (18.24) | 1,477 (51.59) | 2,391 (83.51) | 2,172 (75.86) | 0.83 (0.14) | 1.02 (0.14) | 925 (32.31) |
| LOS_AFR | 2,097 | 39.62 (9.53) | 981 (46.78) | 170.16 (9.03) | 84.6 (21.47) | 1,145 (54.6) | 1,313 (62.61) | 1,381 (65.86) | 0.93 (0.15) | 1.1 (0.15) | 335 (15.98) |
| KCOS_CAU | 2,271 | 51.26 (13.71) | 1,715 (75.52) | 166.37 (8.46) | 75.28 (17.53) | 761 (33.51) | 1,494 (65.79) | 1,711 (75.34) | 0.79 (0.13) | 1.02 (0.15) | 967 (42.58) |
| COS_EAS | 1,569 | 34.52 (13.23) | 798 (50.86) | 162.8 (42.32) | 58.8 (39.23) | 122 (7.78) | 141 (8.99) | 750 (47.8) | 0.81 (0.13) | 0.95 (0.13) | 124 (7.9) |
| MrOS_CAU | 4,587 | 73.97 (5.96) | 0(0) | 174.49 (6.62) | 83.5 (13.07) | 2,862 (62.39) | 2,977 (64.9) | NA | 0.78 (0.12) | 1.07 (0.18) | 684 (14.91) |
| MrOS_AFR | 181 | 72.04 (5.42) | 0 (0) | 174.37 (7.19) | 87.08 (15.76) | 114 (62.98) | 92 (50.83) | NA | 0.87 (0.15) | 1.13 (0.21) | 11 (6.08) |
| MrOS_EAS | 165 | 72.88 (5.04) | 0 (0) | 167.07 (5.98) | 70.29 (8.61) | 88 (53.33) | 75 (45.45) | NA | 0.75 (0.11) | 1.04 (0.18) | 20 (12.12) |
| MrOS_HIS | 111 | 71.84 (4.44) | 0 (0) | 170.39 (5.99) | 81.83 (12.51) | 67 (60.36) | 84 (75.68) | NA | 0.8 (0.12) | 1.04 (0.18) | 12 (10.81) |
| WHI_AFR | 671 | 60.52 (6.76) | 671 (100) | 162.33 (5.52) | 82.49 (17.06) | NA | NA | NA | 0.83 (0.13) | 1.05 (0.17) | 164 (24.44) |
| WHI_HIS | 393 | 60.04 (7.07) | 393 (100) | 157.79 (5.46) | 73.47 (14.86) | NA | NA | NA | 0.73 (0.11) | 0.97 (0.15) | 108 (27.48) |
| CHS_CAU | 432 | 71.06 (4.58) | 131 (30.32) | 169.32 (8.6) | 170.41 (27.6) | 257 (59.49) | 313 (72.45) | 408 (94.44) | 0.77 (0.12) | 1.15 (0.23) | 75 (17.36) |
| CHS_AFR | 190 | 71.63 (4.54) | 92 (48.42) | 167.51 (8.87) | 177.9 (29.53) | 110 (57.89) | 84 (44.21) | 176 (92.63) | 0.84 (0.13) | 1.19 (0.23) | 38 (20) |

**FNK-BMD:** Bone mineral density at femoral neck.

**SPN-BMD:** Bone mineral density at lumbar spine.

**"NA"** indicates that the covariate was not available at that data set.

**"SD"** indicates standard deviation.

**"UKBB"** indicates the UK biobank.

**"LOS"** indicates the Louisiana Osteoporosis Study.

**"KCOS"** indicates the Kansas City Osteoporosis Study.

**"COS"** indicates the China Osteoporosis Study.

**"MrOS"** indicates the Osteoporotic Fractures in Men Study.

**"WHI"** indicates the Women's Health Initiative Clinical Trial and Observational Study.

**"CHS"** indicates the Cardiovascular Health Study.

**"AFR"** indicates African American.

**"CAU"** indicates Caucasian.

**"EAS"** indicates East Asian.

**"HIS"** indicates Hispanic/Latino.

S5 Fig. As expected, the data set originating from the European ancestry population demonstrated a notably lower GD compared to data sets representing other racial groups. Although the first 2 genetic PCs of these racial groups exhibited distinct clustering patterns based on ancestry, no clear demarcation between these categories was evident. In line with previous research findings [52,53], human diversity is observed along a genetic ancestry continuum, devoid of well-defined clusters.

## Variance explained of prediction models in the UKBB Model Selection set

The performance of each model in UKBB Model Selection set is shown in Fig 2. For the regression model with only clinical factors, the $R^2$ of the FNK-BMD and SPN-BMD were 25.388% with a 95% confidence interval (CI) of [25.386%, 25.390%] and 45.313% (95% CI [45.312%, 45.314%]), respectively. For the prediction models including both clinical factors and genetic variants, the performance of the models improved moderately when we set the $P$-value threshold from $5 \times 10^{-8}$ to $5 \times 10^{-6}$. However, when we continued to increase the threshold, the performance of the models deteriorated due to overfitting. The prediction models with the best performance were selected as the prediction models for the follow-up analysis. For FNK-BMD, the prediction model was trained with LASSO when the threshold was set at $5 \times 10^{-6}$ which resulted in an $R^2$ of 27.70% (95% CI [27.56%, 27.84%]), indicating that the prediction model could explain 27.7% of the variance in FNK-BMD in the UKBB British Model Selection set. Compared with the regression model with only clinical factors, the LASSO model for FNK-BMD improved by 2.3% in $R^2$. For SPN-BMD, the prediction model was trained with LASSO when set the threshold as $5 \times 10^{-7}$, which resulted in an $R^2$ of 48.28% (95% CI [48.23%, 48.34%]), indicating that the prediction model could explain 48.28% of the variance in SPN-BMD. Compared with the regression model with only clinical factors, the LASSO model for SPN-BMD improved by 3% in $R^2$.

## Performance of prediction models in the UKBB testing data sets

We applied the prediction models with the highest $R^2$ to predict the BMD values and calculated the $R^2$ and PCC between the predicted and actual BMD values in each test set. As shown in Table 2 and Fig 3, the models for predicting FNK-BMD and SPN-BMD performed similarly in the UKBB British Test set and the UKBB other White set. The model for predicting FNK-BMD explained 24.03% (95% CI [23.89%, 24.17%]) and 24.06% (95% CI [23.83%, 24.29%]) of variance in the UKBB British Test set and the UKBB other White set, respectively. The model for predicting SPN-BMD explained 44.79% (95% CI [44.74%, 44.84%]) and 44.17% (95% CI [44.08%, 44.25%]) of variance, respectively. However, the model performance dropped significantly in the UKBB African ancestry set, East Asian ancestry set, and other ancestry set. Especially among the UKBB African ancestry set, the $R^2$ for FNK-BMD and SPN-BMD were −0.089 (95% CI [−0.173, −0.005]) and −0.333 (95% CI [−0.446, −0.221]), respectively, indicating that the models cannot capture any BMD variability. As suggested by some previous research, the performance decline might result at least partially from decreased similarity in genetic profile [35]. Additionally, for each type of models, we selected the one with the highest $R^2$ from the UKBB Model selection set for testing across all test data sets. The detailed outcomes are presented in S3 and S4 Tables. Similar to the LASSO-based models, these models demonstrated better performance in the in the UKBB British Test set and the UKBB other White set, while the performance varied significantly across other populations.

## Association between predicted BMDs and fragility fracture in the UKBB Fracture case-control set

To examine the relationship between the predicted BMDs and fragility fracture risk, we first conducted a *T* test to compare the mean difference of the predicted BMDs between the fragility

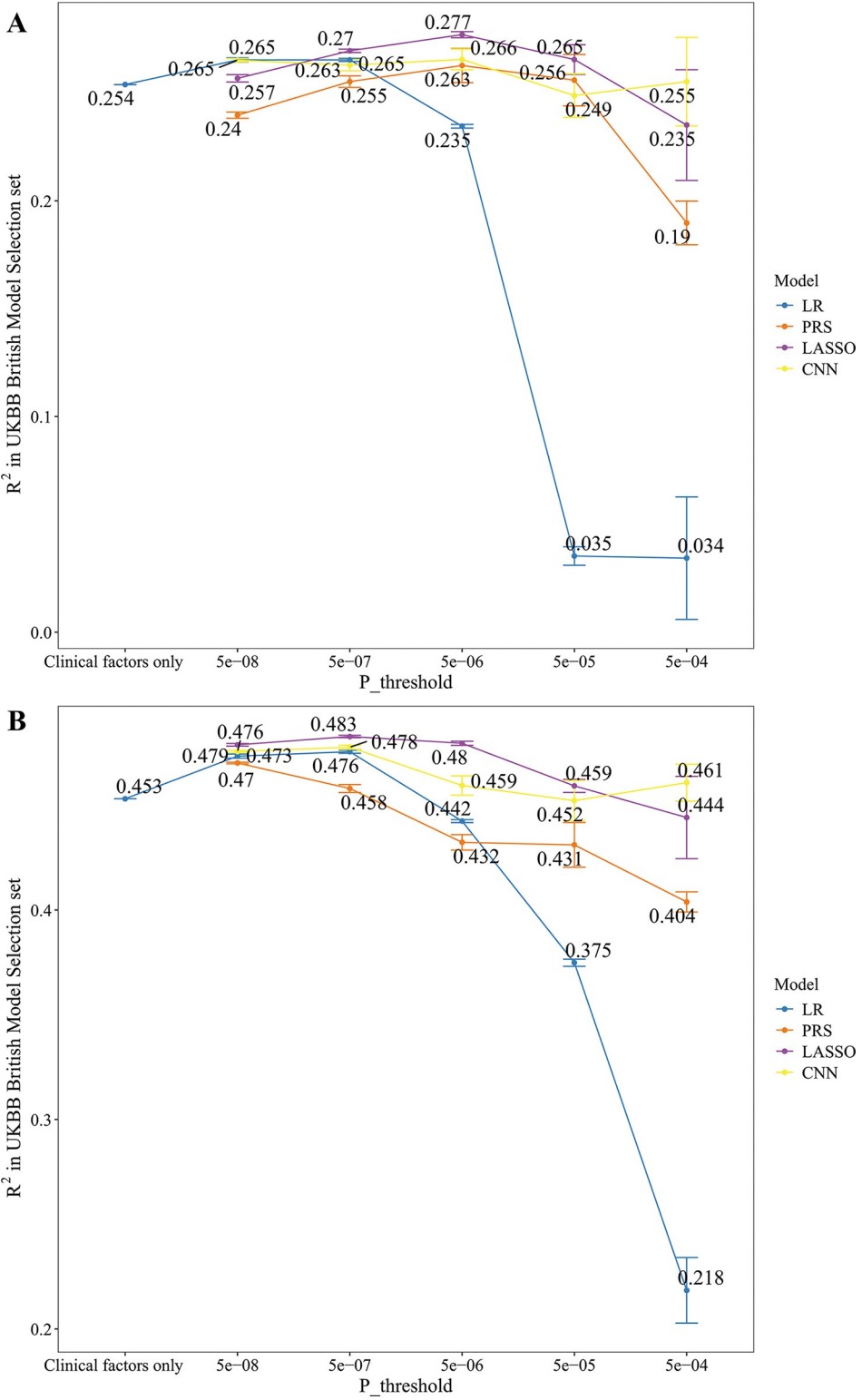

**Fig 2. The performance of each model in UKBB British Model Selection set.** The $R^2$ of each prediction model for FNK-BMD (**A**) and SPN-BMD (**B**) in the UKBB British Model Selection set, respectively. For prediction models including both clinical factors and genetic variants, the performance of the models improved moderately when we set the $P$-value threshold from $5\times10^{-8}$ to $5\times10^{-6}$. However, when we continued to increase the threshold, the performance of the models deteriorated due to overfitting. The prediction models with the best performance were selected as the

prediction models for the follow-up analysis. For FNK-BMD, the prediction model was trained with LASSO when the threshold was set at $5 \times 10^{-6}$. For SPN-BMD, the prediction model was trained with LASSO when set the threshold as $5 \times 10^{-7}$. The vertical short lines represent the upper and lower bounds of the 95% CI. **FNK-BMD:** Bone mineral density at femoral neck. **SPN-BMD:** Bone mineral density at lumbar spine. **$R^2$:** The coefficient of determination. **LR:** Linear regression. **PRS:** polygenic risk score. **LASSO:** Regression with Least absolute shrinkage and selection operator. **CNN:** Convolutional neural network.

fracture cases and the controls in the UKBB fracture case-control set. As expected, the fragility fracture cases had significantly lower mean predicted FNK-BMD and SPN-BMD than the controls (*T* test *P* < 0.001) (Fig 4A). Furthermore, we stratified the population into different groups based on the predicted BMD values and found that a lower predicted BMD was associated with a higher prevalence of fragility fracture (Fig 4B and 4C). For example, among individuals with a predicted FNK-BMD below the fifth percentile, 1,713 out of 14,360 (11.93%) experienced a fragility fracture within the next 10 years. Similarly, among those with a predicted SPN-BMD below the fifth percentile, 1,557 out of 14,360 (10.84%) experienced a fragility fracture. In contrast, the prevalence of fragility fractures was significantly lower among individuals with a predicted BMD above the 95th percentile, with only 3.84% (551 out of 14,360) for FNK-BMD and 3.93% (564 out of 14,360) for SPN-BMD. These results supported that both the predicted FNK-BMD and SPN-BMD had a negative correlation with fracture risk.

## The predictive power of the predicted BMDs for fragility fracture risk

To further evaluate the predictive power of the predicted BMDs for fragility fracture risk, we first conducted a univariate survival analysis with the Kaplan–Meier model to estimate the 10-year cumulative incidence. The log-Rank test showed a significant difference in the cumulative incidence of fragility fracture among different BMD groups (*P* < 0.001; Fig 5A and 5B). Next, we integrated the predicted BMDs and clinical factors to perform a multivariate survival analysis with cox regression to predict the fracture risk of the participants in the next 10 years. As expected, the predicted FNK-BMD and SPN-BMD were both significant in the cox model (*P* < 0.001). The HRs of the predicted FNK-BMD and SPN-BMD were 0.83 (95% CI [0.79, 0.88], corresponding to a 1.44% difference in 10-year absolute risk per standard deviation of BMD) and 0.72 (95% CI [0.68, 0.76], corresponding to a 1.64% difference in 10-year absolute risk per standard deviation of BMD), respectively, which means that for every increase of one standard deviation in BMD, the fracture risk will decrease by 17% and 28%, respectively (Fig 5D). Compared with using only the clinical risk factors, combining the predicted BMDs with the clinical risk factors significantly improved the risk prediction for fragility fracture (likelihood ratio test *p*-values <0.001; Fig 5C).

## Prediction models varied across the independent data sets

The performance of BMD predictive models significantly deteriorated across all the independent data sets, as shown in Table 2. For FNK-BMD, the model performed best in the LOS_-CAU, with an $R^2$ of −0.250 (95% CI [−0.624, 0.125]). For SPN-BMD, the model performed best in the MrOS_CAU, with an $R^2$ of −0.657 (95% CI [−0.878, −0.436]). However, the $R^2$ were less than 0 for both FNK-BMD and SPN-BMD prediction models in all the independent data sets, indicating that the models could not capture any variance of BMD.

Next, we calculated the odds ratios (OR) using the logistic regression to assess the association between the predicted BMDs and fragility fracture risk. As shown in Table 3, the OR between predicted FNK-BMD and fragility fracture was 0.575 (95% CI [0.414, 0.797]) (*P*-

**Table 2. Performance of the BMD prediction models in the testing sets.**

| Data set | | Avg GD | FNK-BMD | | | | | | SPN-BMD | | | | | |
|---|---|---|---|---|---|---|---|---|---|---|---|---|---|---|
| | | | R² | Lower bound | Upper bound | PCC | Lower bound | Upper bound | R² | Lower bound | Upper bound | PCC | Lower bound | Upper bound |
| **UKBB** | Britsh Testing | 0.026 | 0.240 | 0.239 | 0.242 | 0.493 | 0.492 | 0.494 | 0.448 | 0.447 | 0.448 | 0.670 | 0.670 | 0.670 |
| | Other White | 0.043 | 0.241 | 0.238 | 0.243 | 0.501 | 0.500 | 0.502 | 0.442 | 0.441 | 0.443 | 0.669 | 0.668 | 0.669 |
| | EAS | 0.274 | 0.117 | 0.088 | 0.147 | 0.505 | 0.498 | 0.513 | 0.215 | 0.197 | 0.234 | 0.606 | 0.600 | 0.611 |
| | AFR | 0.352 | −0.089 | −0.173 | −0.005 | 0.411 | 0.392 | 0.430 | −0.333 | −0.446 | −0.221 | 0.408 | 0.395 | 0.421 |
| | Other Ancestry | 0.140 | 0.175 | 0.166 | 0.185 | 0.465 | 0.462 | 0.468 | 0.310 | 0.302 | 0.319 | 0.598 | 0.594 | 0.602 |
| **Independent cohorts** | LOS AFR | 0.346 | −0.420 | −0.660 | −0.180 | 0.431 | 0.414 | 0.447 | −1.830 | −2.145 | −1.515 | 0.220 | 0.214 | 0.227 |
| | LOS CAU | 0.047 | −0.250 | −0.624 | 0.125 | 0.564 | 0.559 | 0.568 | −1.641 | −2.046 | −1.236 | 0.302 | 0.295 | 0.309 |
| | COS EAS | 0.274 | −1.866 | −2.553 | −1.178 | 0.264 | 0.262 | 0.266 | −4.471 | −4.990 | −3.952 | 0.161 | 0.154 | 0.168 |
| | KCOS CAU | 0.033 | −0.423 | −1.178 | 0.332 | 0.600 | 0.598 | 0.603 | −1.202 | −1.552 | −0.853 | 0.340 | 0.331 | 0.348 |
| | MrOS AFR | 0.329 | −0.985 | −1.705 | −0.266 | 0.340 | 0.322 | 0.357 | −0.907 | −1.066 | −0.748 | 0.058 | 0.035 | 0.081 |
| | MrOS CAU | 0.042 | −0.762 | −1.640 | 0.116 | 0.377 | 0.374 | 0.380 | −0.657 | −0.878 | −0.436 | 0.207 | 0.198 | 0.215 |
| | MrOS EAS | 0.278 | −1.131 | −2.148 | −0.114 | 0.264 | 0.256 | 0.272 | −0.725 | −0.946 | −0.505 | 0.202 | 0.188 | 0.215 |
| | MrOS HIS | 0.120 | −0.957 | −1.849 | −0.065 | 0.163 | 0.154 | 0.171 | −0.804 | −1.045 | −0.562 | 0.161 | 0.141 | 0.181 |
| | CHS AFR | 0.236 | −3.136 | −3.720 | −2.551 | 0.412 | 0.401 | 0.423 | −3.937 | −4.583 | −3.291 | 0.320 | 0.308 | 0.332 |
| | CHS CAU | 0.065 | −4.892 | −5.613 | −4.170 | 0.437 | 0.434 | 0.440 | −3.772 | −4.374 | −3.170 | 0.282 | 0.278 | 0.286 |
| | WHI AFR | 0.326 | −1.000 | −1.840 | −0.161 | 0.435 | 0.415 | 0.455 | −1.278 | −1.535 | −1.022 | 0.248 | 0.237 | 0.259 |
| | WHI HIS | 0.120 | −0.827 | −1.817 | 0.163 | 0.542 | 0.537 | 0.547 | −1.005 | −1.307 | −0.702 | 0.322 | 0.312 | 0.332 |

**FNK-BMD:** Bone mineral density at femoral neck.

**SPN-BMD:** Bone mineral density at lumbar spine.

**AvgGD:** The average Genetic distance of the individuals in the testing set to the training set.

**R²:** The coefficient of determination.

**PCC:** Pearson correlation coefficient.

**Lower bound:** Lower bound of the 95% confidence interval.

**Upper bound:** Upper bound of the 95% confidence interval.

**"UKBB"** indicates the UK biobank.

**"LOS"** indicates the Louisiana Osteoporosis Study.

**"KCOS"** indicates the Kansas City Osteoporosis Study.

**"COS"** indicates the China Osteoporosis Study.

**"MrOS"** indicates the Osteoporotic Fractures in Men Study.

**"WHI"** indicates the Women's Health Initiative Clinical Trial and Observational Study.

**"CHS"** indicates the Cardiovascular Health Study.

**"AFR"** indicates African American.

**"CAU"** indicates Caucasian.

**"EAS"** indicates East Asian.

**"HIS"** indicates Hispanic/Latino.

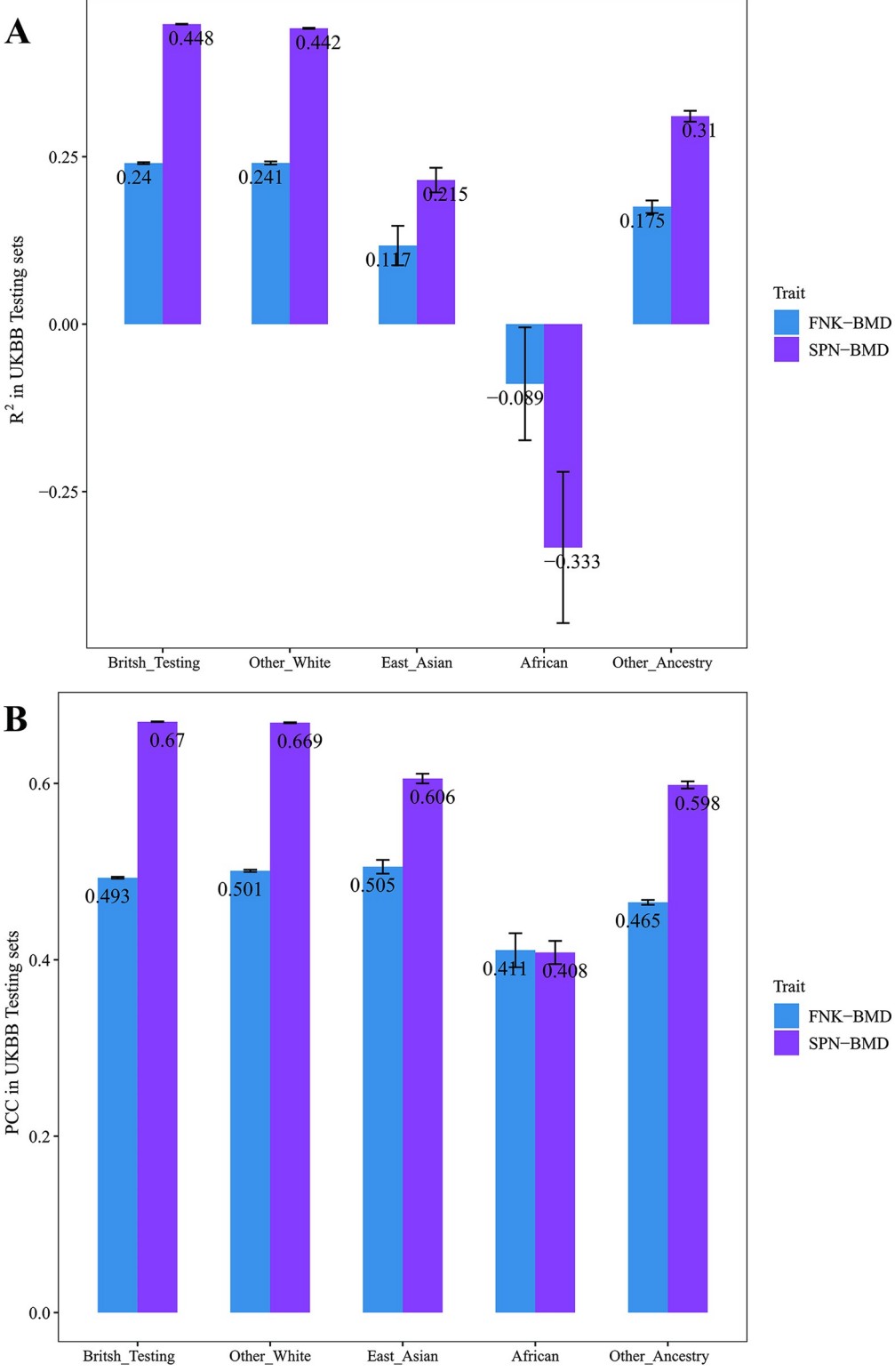

**Fig 3. The predictive performance of each model in UKBB Testing sets. (A)** The $R^2$ between the predicted and actual BMDs in the UKBB British testing set, the UKBB other White set, and the UKBB other Ancestry set. **(B)** The PCC between the predicted and actual BMDs in the UKBB British testing set, the UKBB other White set, and the UKBB other Ancestry set. The models for predicting FNK-BMD and SPN-BMD performed similarly in the UKBB British Test set and the UKBB other White set. However, the model performance drops significantly in the UKBB

other Ancestry set. The vertical short lines represent the upper and lower bounds of the 95% CI. **FNK-BMD:** Bone mineral density at femoral neck. **SPN-BMD:** Bone mineral density at lumbar spine. **R$^2$**: The coefficient of determination. **PCC**: Pearson correlation coefficient.

value = 0.001) in the KCOS_CAU set, indicating that for every one standard deviation increase in BMD, the risk of fracture is reduced by 42.5%. The OR between predicted SPN-BMD and fragility fracture was 0.741 (95% CI 0.589–0.930) (*P*-value = 0.010) in the KCOS_CAU set, indicating that for every one standard deviation increase in BMD, the risk of fracture is

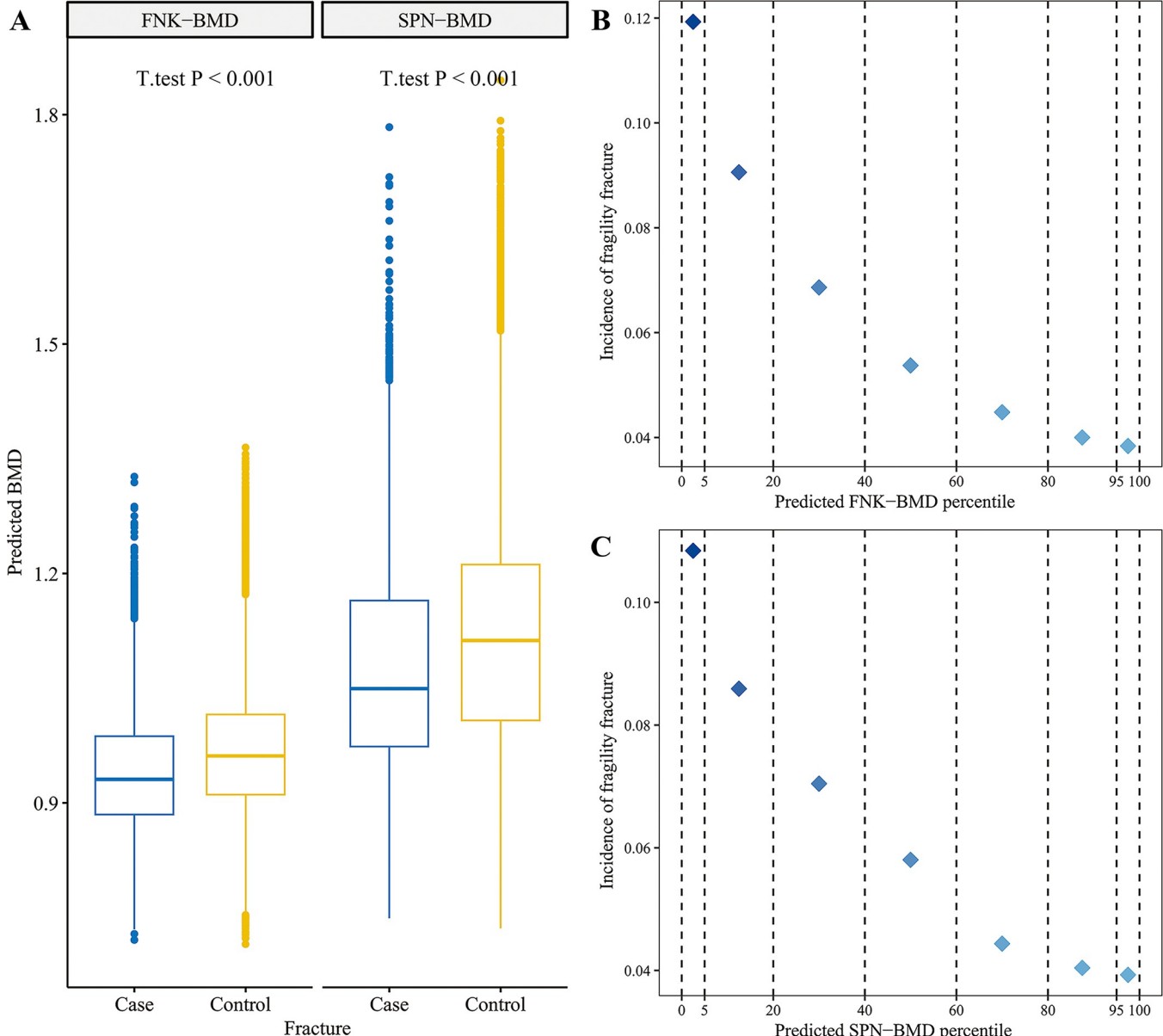

**Fig 4. Association between predicted BMDs and fragility fracture in UKBB Fracture case-control set. (A)** *T* test showed that the fragility fracture cases had significantly lower mean predicted FNK-BMD and SPN-BMD than the controls in the UKBB fracture case-control set. **(B, C)** By stratifying the population into different groups based on the predicted BMD values, we found that a lower predicted BMD was associated with a higher prevalence of fragility fracture. **FNK-BMD:** Bone mineral density at femoral neck. **SPN-BMD:** Bone mineral density at lumbar spine.

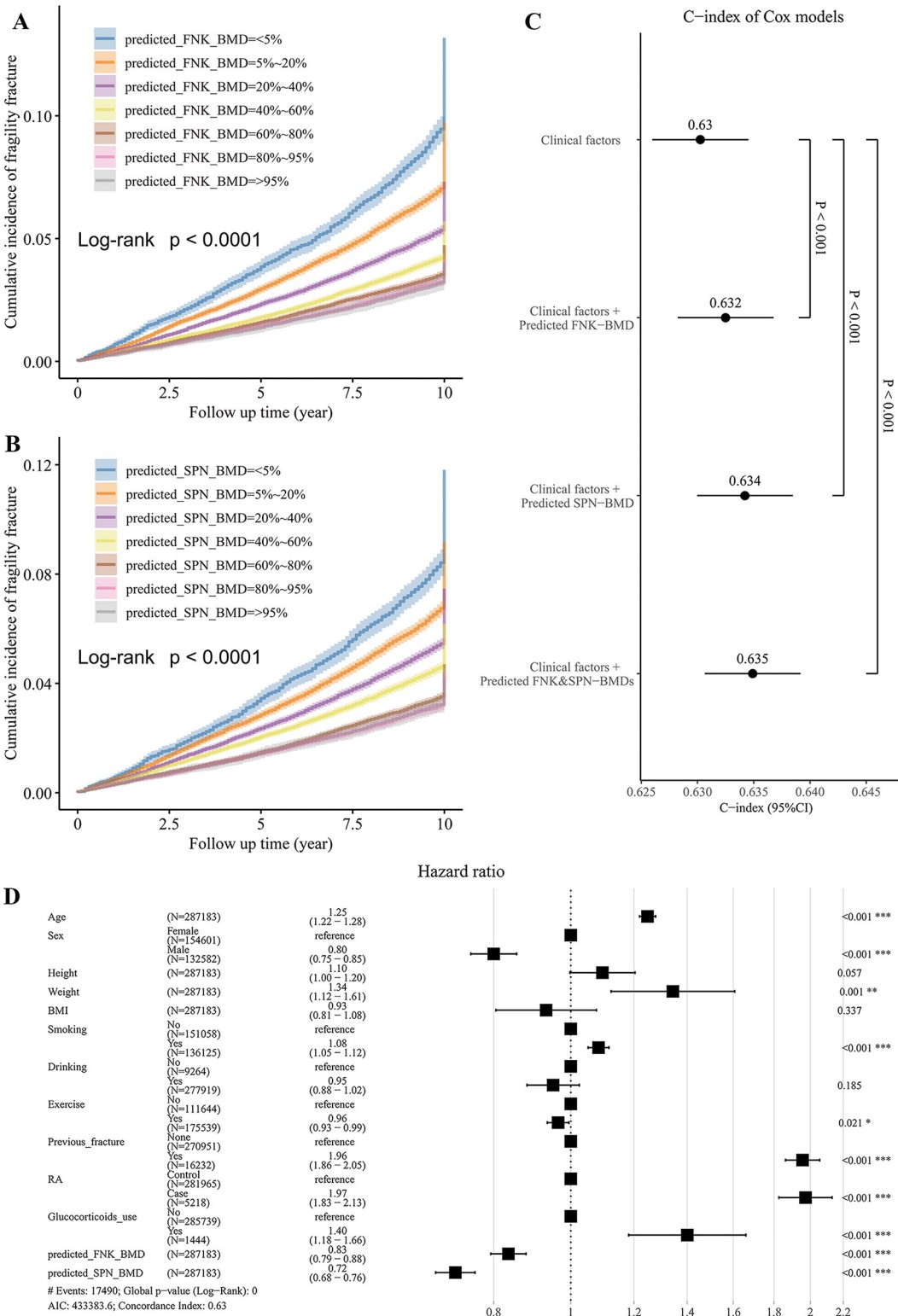

**Fig 5. The strength of predicted BMDs as the predictors of fragility fracture. (A, B)** The univariate survival analysis showed a significant difference in the cumulative incidence of fragility fracture among different BMD groups (censored at 10 years). **(C)** Compared with using only the clinical risk factors, combining the predicted BMDs with the clinical risk factors significantly improved the risk prediction for fragility fracture. **(D)** The predicted FNK-BMD and SPN-BMD were both significant in the cox model. For every increase of one standard deviation in BMD, the fracture risk will decrease by 17% and 28%, respectively. **FNK-BMD:** Bone mineral density at femoral neck. **SPN-BMD:** Bone mineral density at lumbar spine.

**Table 3. The association between the predicted BMDs and fragility fracture risk in independent cohorts.**

| Trait | Data set | Fracture, N (%) | OR | Lower bound | Upper bound | P |
|---|---|---|---|---|---|---|
| FNK-BMD | LOS_CAU | 925 (32.31) | 1.087 | 0.823 | 1.437 | 0.557 |
| | LOS_AFR | 335 (15.98) | 0.897 | 0.744 | 1.082 | 0.254 |
| | KCOS_CAU | 967 (42.58) | 0.575 | 0.414 | 0.797 | 0.001* |
| | COS_EAS | 124 (7.9) | 3.069 | 0.622 | 15.264 | 0.169 |
| | MrOS_CAU | 684 (14.91) | 0.962 | 0.779 | 1.188 | 0.717 |
| | MrOS_AFR | 11 (6.08) | 1.870 | 0.630 | 6.217 | 0.279 |
| | MrOS_EAS | 20 (12.12) | 2.149 | 0.808 | 5.858 | 0.127 |
| | MrOS_HIS | 12 (10.81) | 0.861 | 0.274 | 2.996 | 0.803 |
| | WHI_AFR | 164 (24.44) | 1.056 | 0.808 | 1.386 | 0.690 |
| | WHI_HIS | 108 (27.48) | 0.961 | 0.576 | 1.616 | 0.879 |
| | CHS_CAU | 75 (17.36) | 3.252 | 0.944 | 11.368 | 0.063 |
| | CHS_AFR | 38 (20) | 1.820 | 0.813 | 4.259 | 0.154 |
| SPN-BMD | LOS_CAU | 925 (32.31) | 0.911 | 0.737 | 1.125 | 0.385 |
| | LOS_AFR | 335 (15.98) | 1.009 | 0.838 | 1.216 | 0.922 |
| | KCOS_CAU | 967 (42.58) | 0.741 | 0.589 | 0.930 | 0.010* |
| | COS_EAS | 124 (7.9) | 1.183 | 0.439 | 3.123 | 0.737 |
| | MrOS_CAU | 684 (14.91) | 0.839 | 0.736 | 0.957 | 0.009* |
| | MrOS_AFR | 11 (6.08) | 0.602 | 0.256 | 1.347 | 0.223 |
| | MrOS_EAS | 20 (12.12) | 1.462 | 0.735 | 2.936 | 0.275 |
| | MrOS_HIS | 12 (10.81) | 0.776 | 0.259 | 2.309 | 0.646 |
| | WHI_AFR | 164 (24.44) | 1.048 | 0.832 | 1.320 | 0.691 |
| | WHI_HIS | 108 (27.48) | 0.742 | 0.496 | 1.103 | 0.142 |
| | CHS_CAU | 75 (17.36) | 0.814 | 0.335 | 1.957 | 0.646 |
| | CHS_AFR | 38 (20) | 0.521 | 0.228 | 1.175 | 0.116 |

**FNK-BMD:** Bone mineral density at femoral neck.

**SPN-BMD:** Bone mineral density at lumbar spine.

**OR:** Odds ratio.

**Lower bound:** Lower bound of the 95% confidence interval.

**Upper bound:** Upper bound of the 95% confidence interval.

**"LOS"** indicates the Louisiana Osteoporosis Study.

**"KCOS"** indicates the Kansas City Osteoporosis Study.

**"COS"** indicates the China Osteoporosis Study.

**"MrOS"** indicates the Osteoporotic Fractures in Men Study.

**"WHI"** indicates the Women's Health Initiative Clinical Trial and Observational Study.

**"CHS"** indicates the Cardiovascular Health Study.

**"AFR"** indicates African American.

**"CAU"** indicates Caucasian.

**"EAS"** indicates East Asian.

**"HIS"** indicates Hispanic/Latino.

reduced by 25.9%. The OR value between predicted SPN-BMD and fragility fracture was 0.839 (95% CI [0.736, 0.957]) (*P*-value = 0.009) in the MrOS_CAU set, indicating that for every one standard deviation increase in BMD, the risk of fracture was reduced by 16.1%. However, the predicted BMDs did not show a significant association with fracture in the other data sets.

As suggested in a recent study, the decline in performance can be partially explained by GD [35]. In our investigation, we observed a statistically significant but modest correlation between the residuals of predicted and true BMD values (standardized by the standard

                                        

deviation of true BMD values) and individual-level GD for both FNK-BMD (PCC = 0.04, $P$-value <0.001) and SPN-BMD (PCC = 0.18, $P$-value <0.001). Apart from differences in genetic architecture across cohorts, other factors influencing transferability include genotype–environment interactions and population-specific causal variants. To illustrate, we conducted a backward stepwise regression to develop predictive models for covariates related to BMD in each data set. We noted not only variations in the coefficients of covariates across cohorts but also considerable diversity in the combinations of covariates included in the regression models (see S5 and S6 Tables). For example, while age proved to be a significant predictor for FNK-BMD in most data sets, it was not selected in the COS_EAS set, all MrOS sets, and WHI_HIS set. This discrepancy suggests that the contribution of clinical factors to BMD variation differs across cohorts, leading to variations in the performance of prediction models in different cohorts. This consideration is crucial when applying prediction models to other cohorts, emphasizing the need for adjustments based on local population data.

## Discussion

In summary, we built and tested a series of prediction models for FNK-BMD and SPN-BMD separately in 32,999 independent participants with DXA-BMD measurements from UKBB, and finally the LASSO regression was selected as the best prediction models. We observed that integrating the clinical risk factors and genetic variants could slightly improve the predictive performance in the European white population from UKBB. Predicted BMDs for 287,183 European individuals without DXA-BMD measurements were strongly associated with fragility fracture risk. However, the models' performance decreased significantly on 5 UKBB test sets and 12 independent cohorts of diverse ancestries (totaling over 15,000 individuals).

Unlike previous studies that used only GWAS-significant SNPs, we included a broader range of SNPs, improving the variance explained in BMD. For example, a previous study including 62 SNPs associated explained only 1.3% (approximately 42% when adding gender, age, and weight) of the variance in FNK-BMD and 1.6% (approximately 35% when adding gender and weight) of the variance in SPN-BMD [16], while our models explained up to 48% for SPN-BMD. Interestingly, we observed an obvious difference in the explanation in variance of FNK-BMD and SPN-BMD by the prediction model. The prediction model could explain approximately 28% of the variance in FNK-BMD while approximately 48% of the variance in SPN-BMD. This may be due to several reasons: First, the SPN and FNK have different biological and structural characteristics that might influence the accuracy of BMD predictions. Second, the relative contributions of genetic and clinical factors vary across skeletal sites [54]. For example, FNK-BMD is more sensitive to the clinical effects such as physical activity and nutritional intake, while SPN-BMD is more sensitive to body weight [11,55–57]. In our study, we determined that the most suitable $P$-value thresholds for FNK-BMD and SPN-BMD were $5\times10^{-6}$ and $5\times10^{-7}$, encompassing 836 and 276 SNPs, respectively. However, it is crucial to note that the optimal threshold is not universally constant; rather, it depends on the specific trait and the sample size of the training data. For example, a prior study developed a PRS for eBMD involving 341,449 individuals in the training set, and they incorporating 21,717 SNPs with a $P$-value threshold of $5\times10^{-4}$ [17]. Given that obtaining DXA-BMD measurements is more challenging and costly compared to eBMD, our training set was limited to a sample size of 17,964. Naturally, there are likely numerous BMD-associated SNPs have been overlooked. As more DXA-BMD data accumulates and computational power improves, we believe that future studies have the potential to expand the inclusion of selected SNPs, thereby capturing a broader spectrum of BMD variation.

Furthermore, many existing PRS researches only accounted for the heritable component of a trait and ignores the significant role of environmental and lifestyle factors in disease etiology, further limiting their overall prediction power. In contrast, we considered both genetic and clinical factors such as sex, age, height, and weight, which have been proven to be closely associated with BMD [11]. By highlighting the significance of clinical factors, our research provides a more comprehensive understanding of BMD prediction and the assessment of osteoporosis risk.

Another contribution of our study to existing research is that we underscore the imperative for training models on diverse population. As has been widely noted, when a genetic prediction approach is applied to populations of different ancestries, its predictive performance may vary, sometimes to a large extent [58]. Our study further highlights that, the prediction approaches vary in predictive performance not only among populations of different ethnics, but also among populations of similar ethnic but different geographic background. This conclusion was also proposed in a previous PRS study for coronary artery disease (CAD) [59]. It was reported that GD is an important contributor to decay of predictive performance, which corresponds to decreased similarity in genetic profile (assessed by relatedness, linkage disequilibrium, and/or minor allele frequency differences, fixation index (Fst) and so on) between testing individuals and training data [35]. However, GD cannot fully explain the reduced predictive performance, as a decrease in the predictive efficacy was also observed in data sets from other European populations with different geographical backgrounds. Additionally, the absence of certain variants can also affect the accuracy of predictive models. Such rare variants are particularly noticeable because their frequency in the population is too low to be detected. For example, the SNP rs554533790, located in an intergenic region, appears in the 1000 Genomes data set with a minor allele frequency of less than 0.005. In our study, rs554533790 has a minor allele frequency of approximately 0.0004 in the UKBB CAU populations, and zero in the East EAS and AFR populations. However, this variant was not detected in other independent data sets. Although rare, rs554533790 has a high weight in our FNK-BMD predictive model, even higher than age, highlighting its significance. Therefore, the absence of this SNP in other independent data sets will inevitably reduce the model's predictive performance. However, as a rare variant in an intergenic region, the function of rs554533790 remains largely unknown. Further functional or mechanistic studies are needed to elucidate its role in bone. Some clinical factors also contribute to performance degradation, such as age and weight, which were identified as crucial elements in our FNK- and SPN-BMD prediction models. The most striking explanation for these variations is the disparate age distributions and sex ratios present in the data sets. Moreover, differences in diet, physical activity levels, sun exposure, and other lifestyle and environmental factors across populations can result in inconsistent impacts of these clinical factors on BMD, even among individuals from the similar ethnic background. These variances may represent another contributing factor to the diminished predictive capability of our model in different populations. Therefore, every single population-specific risk prediction model needs to be derived on their very own training data set or at least verified for application on the target population, even if the models trained in other populations of the same ethnicity are available.

Strengths of our study include the large sample size from UKBB and comparison of various types of prediction models that integrated of both genetic and clinical factors, which enable us to further enhance the performance of BMD predictions. However, this study has some limitations: First, the distribution of clinical factors varied across the UKBB data sets and the independent cohorts. Second, the included genetic factors included in our model is mainly based on GWAS results, which can only detect SNVs associated with phenotypes, but not other types of variants or gene expression regulation. Besides SNVs, other types of genomic variants, such

as copy number variants (CNVs) and genetic rearrangements (GRs) may also affect BMD [60,61]. These variants may alter gene expression or function by causing gene duplication, deletion, translocation, or inversion. BMD is also affected by the complex gene expression regulation, such as epigenetic modification, transcription factor, and noncoding RNA [11,62]. These factors may regulate gene activity and interaction in different cell types, developmental stages, and environmental conditions. Third, the selection of fragility fracture cases utilizes self-report, and hospital inpatient data for the definition of fragility fracture cases, which may lead to some incorrect or inaccurate classification.

The most significant aspect of our research is demonstrating that this genomic prediction method can be used to forecast the risk of fragility fractures in advance. Though our model did not capture all BMD variance, it is still helpful for assessing fracture risk. We found that the predicted BMDs could improve the performance of fracture prediction over and above that of clinical risk factors alone, such as height, weight, smoking, drinking, and exercise. Currently, DXA-BMD measurement of the spine and hip is the gold standard imaging test for diagnosing osteoporosis and assessing fragility fracture risk [1]. However, due to its high cost, large size of the equipment and requirement for professional operation, DXA measurement is not suitable for large-scale population screening, especially in medically underdeveloped areas. With the development of whole genome sequencing technology, the cost of sequencing a human genome has been significantly reduced, which means that gene testing technology has become more accessible to the public, and can provide strong support for various fields, such as personalized medicine, precision medicine, and preventive medicine [62]. For areas where DXA measurement is inconvenient, obtaining genetic information may be more convenient and cost-effective. Moreover, building prediction models based on genetic information may be particularly useful in this situation, because the one-time cost of genotyping can be used not only for evaluation of osteoporosis but also for prediction of other complex diseases, such as atrial fibrillation, CAD, type 2 diabetes, breast cancer, colorectal cancer, and prostate cancer [21,63–67]. It is imperative to highlight that the predictive models should serve as tools to support, not replace, the clinical judgment of physicians and the discernment of individuals. Decisions regarding clinical examinations or proactive interventions should not be made on the predictive outcomes alone. Vigilance against disease risks must be maintained, even if the model indicates a lower probability of illness. A realistic expectation for our approach is to identify individuals who have a higher risk of osteoporosis or fractures compared to the general populace, thereby encouraging them to recognize potential health threats, embrace positive lifestyle changes, and pursue timely diagnosis and preventive measures.

Of course, before these genomic prediction models can be formally applied in clinical settings, further research is needed. This includes collecting data from diverse populations to build prediction models that benefit a wider range of individuals, developing more complex models that incorporate a broader spectrum of variant types to enhance predictive accuracy, and using interpretable models and mechanistic studies to identify potential therapeutic targets.

In conclusion, our study shows that incorporating genetic factors can enhance DXA-BMD prediction beyond clinical factors alone. Adjusting SNP inclusion thresholds (e.g., $5\times10^{-6}$ or $5\times10^{-7}$) instead of only using GWAS-significant SNPs ($P < 5\times10^{-8}$) can improve model performance, depending on the trait and sample size. The clinical utility of BMD prediction models can help prioritize individuals at high risk in early stages. This early identification encourages individuals to take their risk seriously, participate actively in necessary early screenings, and proactively improve their lifestyle habits. However, this is contingent upon having sufficient data from populations with similar backgrounds to construct accurate predictive models. Our study also emphasizes the importance of prediction model training in

diverse populations, only across populations of different ethnicities but also among populations with similar ethnic backgrounds but different geographic origins.

## Supporting information

**S1 Table. Used ICD10 and self-reported codes for extracting fracture and RA cases. RA,** rheumatoid arthritis.
(XLSX)

**S2 Table. Input features of prediction models. LR,** linear regression; **PRS,** polygenic risk score; **LASSO,** regression with least absolute shrinkage and selection operator; **CNN,** convolutional neural network; **FNK-BMD,** bone mineral density at femoral neck; **SPN-BMD, bone mineral density at lumbar spine.**
(XLSX)

**S3 Table. Performance of all 4 FNK-BMD prediction models in the testing data sets. LR**, linear regression; **PRS**, polygenic risk score; **LASSO**, regression with least absolute shrinkage and selection operator; **CNN**, convolutional neural network; $R^2$, the coefficient of determination; **PCC**, Pearson correlation coefficient.
(XLSX)

**S4 Table. Performance of all 4 SPN-BMD prediction models in the testing data sets. LR**, linear regression; **PRS**, polygenic risk score; **LASSO**, regression with least absolute shrinkage and selection operator; **CNN**, convolutional neural network; $R^2$, the coefficient of determination; **PCC**, Pearson correlation coefficient.
(XLSX)

**S5 Table. Coefficients of covariates for FNK-BMD in each cohort. Coef:** Coefficient of the covariates after stepwise regression. **Pval:** *P*-values of the covariates after stepwise regression. '-' represents that the covariate was removed from the stepwise regression (or not available in this data set).
(XLSX)

**S6 Table. Coefficients of covariates for SPN-BMD in each cohort. Coef:** Coefficient of the covariates after stepwise regression. **Pval:** *P*-values of the covariates after stepwise regression. '-' represents that the covariate was removed from the stepwise regression (or not available in this data set).
(XLSX)

**S1 Fig. The architecture diagram of the CNN model.** The convolutional layers are 1-dimensional ($1 \times 4$) with pooling layers.
(TIF)

**S2 Fig. Selection of λ for LASSO models.** The $R^2$ (coefficient of determination) were calculated within the UKBB Model Selection set.
(TIF)

**S3 Fig. Selection of drop rate for CNN models.** The $R^2$ (coefficient of determination) were calculated within the UKBB Model Selection set.
(TIF)

**S4 Fig. The diagram of the models and the processing of model training, selection, and evaluation. PRS**, polygenic risk score; **LASSO**, regression with least absolute shrinkage and

selection operator; $R^2$, the coefficient of determination; **PCC**, Pearson correlation coefficient.
(TIF)

**S5 Fig. The first 2 genetic PCs of all individuals and the GD to the training data.** PC, principal component of genetic profile; GD, individual-level genetic distance.
(TIF)

**S1 Appendix. Supplementary methods.**
(DOCX)

## Acknowledgments

This research has been conducted using the UK Biobank Resource under project number 63047. We appreciate the generosity of all cohort volunteers. This work was supported in part by the High Performance Computing Center of Central South University.

## Author Contributions

**Conceptualization:** Hong-Mei Xiao, Hong-Wen Deng.

**Data curation:** Yong Liu, Kuan-Jui Su, Anqi Liu, Qing Tian, Lan-Juan Zhao, Chuan Qiu, Zhe Luo, Hui Shen, Hong-Mei Xiao, Hong-Wen Deng.

**Formal analysis:** Yong Liu.

**Funding acquisition:** Hong-Mei Xiao, Hong-Wen Deng.

**Methodology:** Yong Liu, Xiang-He Meng, Chong Wu, Martha I Gonzalez-Ramirez.

**Project administration:** Hong-Wen Deng.

**Software:** Yong Liu.

**Supervision:** Chong Wu, Hong-Mei Xiao, Hong-Wen Deng.

**Validation:** Yong Liu.

**Visualization:** Yong Liu.

**Writing – original draft:** Yong Liu.

**Writing – review & editing:** Xiang-He Meng, Chong Wu, Kuan-Jui Su, Anqi Liu, Qing Tian, Lan-Juan Zhao, Chuan Qiu, Zhe Luo, Martha I Gonzalez-Ramirez, Hui Shen, Hong-Mei Xiao, Hong-Wen Deng.

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
