## [Editor Report · Decision Letter 0]

26 Mar 2024

Dear Dr Liu, 

Thank you for submitting your manuscript entitled "The performance of genetic-enhanced DXA-BMD predicting models trained in UK biobank varies across diverse ethnic and geographical populations" for consideration by PLOS Medicine.

Your manuscript has now been evaluated by the PLOS Medicine editorial staff and I am writing to let you know that we would like to send your submission out for external peer review.

Please re-submit your manuscript on or after April 2nd 2024.

Kind regards,

Pippa

Philippa C. Dodd, MBBS MRCP PhD

PLOS Medicine

pdodd@plos.org

---

## [Decision Letter · Decision Letter 1]

17 May 2024

Dear Dr. Liu,

Many thanks for submitting your manuscript "The performance of genetic-enhanced DXA-BMD predicting models trained in UK biobank varies across diverse ethnic and geographical populations, PMEDICINE-D-24-00964R1” to PLOS Medicine. The paper has been reviewed by two subject experts and a statistician; their comments are included below and can also be accessed here: 

[LINK]

As you will see, the reviewers were positive about the paper but, they raised a number of questions about specific study details and the methodological approach. After discussing the paper with the editorial team and an academic editor with relevant expertise, I’m pleased to invite you to revise the paper in response to the reviewers’ comments. We plan to send the revised paper to some of all of the original reviewers*, and of course we cannot provide any guarantees at this stage regarding publication.

When you upload your revision, please include a point-by-point response that addresses all of the reviewer and editorial points, indicating the changes made in the manuscript and either an excerpt of the revised text or the location (eg: page and line number) where each change can be found. Please submit a clean version of the paper as the main article file and a version with changes marked should as a marked-up manuscript. Please also check the guidelines for revised papers at http://journals.plos.org/plosmedicine/s/revising-your-manuscript for any that apply to your paper.

We ask that you submit your revision by June 7th 2024. However, if this deadline is not feasible, please contact me by email, and we can discuss a suitable alternative.

Please don’t hesitate to contact me directly with any questions (pdodd@plos.org). If you reply directly to this message, please be sure to ‘Reply All’ so your message comes directly to my inbox.

Kind regards,

Pippa

Philippa Dodd MBBS MRCP PhD

PLOS Medicine

plosmedicine.org

pdodd@plos.org

*Please note: If your article is accepted, you may have the opportunity to make the peer review history publicly available. The record will include editor decision letters (with reviews) and your responses to reviewer comments. If eligible, we will contact you to opt in or out.

Editorial comments:

1) The editorial team are in agreement that your manuscript is very interesting and well presented. Overall, opinions were somewhat split because of the limited direct clinical application. This was echoed by the Academic Editor (please see below). Please consider this point as part of your revisions and ensure to include clear discussion, not only of the potential usefulness of such a model, but the benefits and barriers to its implementation in clinical practice.

2) Data Availability

PLOS Medicine requires that the de-identified data underlying the specific results in a published article be made available, without restrictions on access, in a public repository or as Supporting Information at the time of article publication, provided it is legal and ethical to do so. Please see the policy at http://journals.plos.org/plosmedicine/s/data-availability and FAQs at 

http://journals.plos.org/plosmedicine/s/data-availability#loc-faqs-for-data-policy

The Data Availability Statement (DAS) requires revision. For each data source used in your study: 

3) Of all authors who submit modelling studies we ask that the following points are included in the main manuscript. Please review the list below and ensure that each item is included:

* Please provide a diagram that shows the model structure, including how the disease natural history is represented, the process and determinants of disease acquisition, and how the putative intervention could affect the system.

* Please provide a complete list of model parameters, including clear and precise descriptions of [the meaning of each parameter, together with the values or ranges for each, with justification or the primary source cited, and important caveats about the use of these values noted].

* Please provide a clear statement about how the model was fitted to the data [including goodness-of-fit measure, the numerical algorithm used, which parameter varied, constraints imposed on parameter values, and starting conditions].

* For uncertainty analyses, please state the sources of uncertainties quantified and not quantified [can include parameter, data, and model structure].

* Please provide sensitivity analyses to identify which parameter values are most important in the model. Uncertainty estimates seek to derive a range of credible results on the basis of an exploration of the range of reasonable parameter values. The choice of method should be presented and justified.

* Please discuss the scientific rationale for this choice of model structure and identify points where this choice could influence conclusions drawn. Please also describe the strength of the scientific basis underlying the key model assumptions.

Items are derived from Geoffrey P Garnett, Simon Cousens, Timothy B Hallett, Richard Steketee, Neff Walker. Mathematical models in the evaluation of health programmes. (2011) Lancet DOI:10.1016/S0140-6736(10)61505-X

4) Statistical reporting

Throughout, including tables and figures, please quantify the main results with 95% CIs and p values.

When reporting p values please report as <0.001 and where higher as p=0.002, for example. If not reporting p values, for the purpose of transparent data reporting, please clearly state the reasons why not. When reporting 95% CIs please separate upper and lower bounds with commas instead of hyphens as the latter can be confused with reporting of negative values.

Please include the actual amounts and/or absolute risk(s) of relevant outcomes (including NNT or NNH where appropriate), not just relative risks or correlation coefficients. (example for absolute risks: PMID: 28399126).

5) Abstract layout

Please ensure you structure your abstract using the PLOS Medicine headings (Background, Methods and Findings, Conclusions). Please combine the Methods and Findings sections into one section, “Methods and findings”.

6) Author summary

At this stage, we ask that you include a short, non-technical Author Summary of your research to make findings accessible to a wide audience that includes both scientists and non-scientists. The authors summary should consist of 2-3 succinct bullet points under each of the following headings:

• Why Was This Study Done? Authors should reflect on what was known about the topic before the research was published and why the research was needed.

• What Did the Researchers Do and Find? Authors should briefly describe the study design that was used and the study’s major findings. Do include the headline numbers from the study, such as the sample size and key findings. 

• What Do These Findings Mean? Authors should reflect on the new knowledge generated by the research and the implications for practice, research, policy, or public health. Authors should also consider how the interpretation of the study’s findings may be affected by the study limitations. In the final bullet point of ‘What Do These Findings Mean?’, please describe the main limitations of the study in non-technical language.

The Author Summary should immediately follow the Abstract in your revised manuscript. This text is subject to editorial change and should be distinct from the scientific abstract. Please see our author guidelines for more information: https://journals.plos.org/plosmedicine/s/revising-your-manuscript#loc-author-summary

7) Introduction layout

Please address past research and explain the need for and potential importance of your study. Indicate whether your study is novel and how you determined that. If there has been a systematic review of the evidence related to your study (or you have conducted one), please refer to and reference that review and indicate whether it supports the need for your study.

8) Discussion layout

Please present and organize the Discussion as follows: a short, clear summary of the article's findings; what the study adds to existing research and where and why the results may differ from previous research; strengths and limitations of the study; implications and next steps for research, clinical practice, and/or public policy; one-paragraph conclusion.

Comments from the reviewers:

Reviewer #1: The authors present a very interesting manuscript on the use of predictive modelling combining not just genetic data, but also clinical data. This exercise is highly warranted and presents a more viable and cost-effective approach to evaluating the risk of osteoporosis and fracture susceptibility. 

A few comments are included below:

- How were clinical risk factors chosen for BMD and fracture risk?

- Was LS BMD available for L1-L4 or L2-L4?

- Could the low predictive modelling be due to the utilisation of GWAS data and inclusion of common variants? 

- Was there any particular clinical risk factor(s) included in the analysis that was driving the association? Perhaps even SNP(s) or Indels? In the case of the latter, in which genes did these fall? 

- Is there a reason why predictive modelling in the UKBB Training sets seemed to have worked better for spine compared to FN BMD?

- In the case of WGS, were variants annotated? If yes, which tool was used? Was allele balance accounted for? What was the coverage cut-off used for the variants?

Reviewer #2: "The performance of genetic-enhanced DXA-BMD predicting models trained in UK biobank varies across diverse ethnic and geographical populations" reports the outcome of developing various prediction models, towards early detection of osteoporosis in the neck and spine. The focus on genetic data is in response to DXA-based bone mineral density measurement (BMD) being relatively inaccessible, and possibly inappropriate for general screening due to proportion of negative cases. The models were developed on largely white individuals between 40-69 years from the UKBB dataset, and validated on a number of independent cohorts of varying ethnicity distributions. It was found that while prediction was improved by the inclusion of genetic data, performance on non-white ethnicity cohorts was significantly poorer than on the white demographic as used to develop the prediction models.

This paper presents justified support for the integration of suitable genetic data in prediction models for osteoporosis screening, and raises timely concerns about such models possibly not being directly applicable to demographics outside that which were used in their development. A number of issues might be considered:

1. In Line 154, it is stated that "Then, we selected the model with best performance and assessed the predictive performance in three UKBB test sets and 12 independent cohorts". It might be clarified whether this "model with best performance" is separately selected for the FNK and SPN classes, or otherwise.

2. Related to the above, the best model (from the four) might be indicated for the main results in Table 2. Moreover, the full results for all four models on both tasks might be reported if possible.

3. In Line 171, it is stated that family relationship inference was performed using KING software. The reliability of such inference might be briefly stated.

4. In Line 172, it is stated that only individuals with no relative 3rd degree or closer were retained, to ensure sample independence. It might be clarified as to what was done when pairs (or groups) of individuals with close relatives were found. Were all of them excluded, or were one of such pairs (or groups) retained?

5. In the Phenotype measurements and quality control section, the treatment of missing variables (if any; i.e. imputation, exclusion, etc.) might be briefly stated, if relevant.

6. In Line 245, the 1/1000 percentile is referred to. It might be clarified as to how this threshold was chosen. Also, it might be confirmed as to whether this is 1/1000 (i.e. 0.1th percentile), or indeed the 0.001th percentile.

7. In Line 296, it is stated that the most significant SNP was chosen within each LD window. Was there any reason to restrict each LD window to a single SNP?

8. In Line 317, a CNN model is described (also in S1 Fig). It might be clarified as to the input dimensions of the model - is it 1D or 2D, and if the latter, what would be the width & height dimensions of the input?

9. A number of minor grammatical/phrasing issues might be considered, e.g.

(Line 231) "in other ethnic population" -> "populations"

(Line 235) "The rest participants" -> "The rest of the participants"

(Line 294) "To reduce of the input features" -> "To reduce the input features"

etc.

Reviewer #3: In this study, the authors developed a predictive model for dual-energy X-ray absorptiometry (DXA)bone mineral density (BMD) at femoral neck (FNK) and lumbar spine (SPN), based on genetic and clinical factors within a Caucasian cohort from the UK Biobank (UKBB). The model was then tested across other ancestry groups within the UKBB and several independent cohorts. The findings revealed that the predictive model performed well within the UKBB Caucasian population, but there was a notable variance in performance across other populations and independent cohorts. The study demonstrated that genetic factors could improve the performance of DXA-BMD prediction beyond that of clinical factors alone. Moreover, the predicted BMDs significantly improved the prediction of fracture risk. This study provides new insights into predicting osteoporosis and fracture risk, aiding clinicians in better understanding and preventing fragile fractures, thereby improving patient care and quality of life. Additionally, testing the BMD predictive model's performance across different ethnic backgrounds underscores the importance of training models on diverse population datasets, which is scientifically substantiated for public health measures targeting prevention. I have several comments for authors to consider improve the manuscript:

1. To underscore the potential clinical and research implications of this study, it would be beneficial to expand upon the significance of the findings in the Conclusion section. This could include discussing how their findings might influence clinical practices or directions for futures tudies.

2. The authors used linear regressions to demonstrate that the impact of various clinical factors on BMD prediction differs across cohorts, even within the same ethnicity (Line 489-497). A more detailed exploration of the underlying reasons for these differences would be informative in the Discussion.

3. The study grouped all non-Caucasian ethnic populations within the UKBB into a into a single dataset (Line 229-241). For a more nuanced analysis, it would be advantageous todelineate these populations byspecific ethnicities. This would enhance the comparability of the model's performance with other ethnic groups in independent cohorts.

4. The inclusion of additional variables such as glucocorticoid use and rheumatoid arthritis status in the Cox proportional hazards regression analysis (Lines 359-362) should be justified. Clarifying the rationale for these choices would strengthen the methodological transparency of the study.

5. There are some detail problems in some figures and tables, such as a spelling mistake in Table 2: 'Average_DS' should be corrected to 'Average_GD'; and the full names of abbreviations like LR, PRS, etc., in Fig 2 should be clarified in the legend. A thorough review of all visual elements is recommended to ensure accuracy and clarity.

[LINK]

Comments from the Academic Editor:

I agree with offering major revision. However, I understand the reservations of the editorial team. From a clinician's perspective, I am not sure these findings will have direct clinical implications. To date, there is no routine screening of genetic factors for common diseases such as osteoporosis, since the cost-effectiveness of such procedure is unknown. It would be helpful for the authors to comment in this respect.

1. Please upload any figures associated with your paper as individual TIF or EPS files with 300dpi resolution at resubmission; please read our figure guidelines for more information on our requirements: http://journals.plos.org/plosmedicine/s/figures. While revising your submission, please upload your figure files to the PACE digital diagnostic tool, https://pacev2.apexcovantage.com/. PACE helps ensure that figures meet PLOS requirements. To use PACE, you must first register as a user. Then, login and navigate to the UPLOAD tab, where you will find detailed instructions on how to use the tool. If you encounter any issues or have any questions when using PACE, please email us at PLOSMedicine@plos.org.

To submit your revised manuscript please use the following link:

---

## [Decision Letter · Decision Letter 2]

12 Jul 2024

Dear Dr. Liu,

Thank you very much for re-submitting your manuscript "The performance of genetic-enhanced DXA-BMD predicting models trained in UK biobank varies across diverse ethnic and geographical populations" (PMEDICINE-D-24-00964R2) for review by PLOS Medicine.

I have discussed the paper with my colleagues and the academic editor and it was also seen again by 3 reviewers. I am pleased to say that provided the remaining editorial and production issues are dealt with we are planning to accept the paper for publication in the journal.

[LINK]

We look forward to receiving the revised manuscript by Jul 19 2024 11:59PM.   

Kind regards,

Pippa

Philippa Dodd, MBBS MRCP PhD

Senior Editor 

PLOS Medicine

plosmedicine.org

pdodd@plos.org

Requests from Editors:

GENERAL

Thank you for your responses to previous editor and reviewer requests, please see below for further comments which we require you address in full.

The editorial comments pertain largely to specific content and formatting requirements. Some items may not apply and others may have already been incorporated appropriately but please review the complete list and amend as necessary.

The main manuscript is very long thus makes for rather a challenging read at times. It would benefit from being made more concise throughout, particularly the methods and the discussion sub-sections. Specific comments are detailed below.

DATA AVAILABILITY STATEMENT

Please note that a study author cannot be a contact for data inquiries. The Data Availability Statement (DAS) requires revision. In respect of the ‘raw whole genome array or sequencing (WGS) data’: 

TITLE

Please revise your title according to PLOS Medicine's style. Your title must be nondeclarative and not a question. It should begin with main concept if possible. "Effect of" should be used only if causality can be inferred, i.e., for an RCT. Please place the study design ("A randomized controlled trial," "A retrospective study," "A modelling study," etc.) in the subtitle (ie, after a colon).

ABSTRACT

Please structure your abstract using the PLOS Medicine headings (Background, Methods and Findings, Conclusions).

Please combine the Methods and Findings sections into one section, “Methods and findings”.

Abstract Background: Please provide context of why the study is important. The final sentence should clearly state the study question.

Abstract Background: Please provide context of why the study is important. The final sentence should clearly state the study question.

Abstract Methods and Findings:

Please ensure that all numbers presented in the abstract are present and identical to numbers presented in the main manuscript text.

Please include the study design, population and setting, number of participants, years during which the study took place, length of follow up, and main outcome measures.

Please quantify the main results with 95% CIs and p values.

Please include the important dependent variables that are adjusted for in the analyses.

Please include the actual amounts and/or absolute risk(s) of relevant outcomes (including NNT or NNH where appropriate), not just relative risks or correlation coefficients. (example for absolute risks: PMID: 28399126). 

Please include a summary of adverse events if these were assessed in the study.

In the last sentence of the Abstract Methods and Findings section, please describe the main limitation(s) of the study's methodology.

Abstract Conclusions:

Please address the study implications without overreaching what can be concluded from the data; the phrase "In this study, we observed ..." may be useful.

Please interpret the study based on the results presented in the abstract, emphasizing what is new without overstating your conclusions.

Please avoid vague statements such as "these results have major implications for policy/clinical care". Mention only specific implications substantiated by the results.

Please avoid assertions of primacy ("We report for the first time....")

STATISTICAL REPORTING 

Previously we asked, “Please include the actual amounts and/or absolute risk(s) of relevant outcomes (including NNT or NNH where appropriate), not just relative risks or correlation coefficients. (example for absolute risks: PMID: 28399126).” We did not see any amendments related to this comment or any specific response rebuttal. Throughout, including the abstract, please report measurements of actual risk, this is a prerequisite to publication.

AUTHOR SUMMARY

Thank you for including an author summary which requires some revision. The authors summary should consist of 2-3 succinct bullet points under each of the following headings:

• Why Was This Study Done? Authors should reflect on what was known about the topic before the research was published and why the research was needed.

• What Did the Researchers Do and Find? Authors should briefly describe the study design that was used and the study’s major findings. Do include the headline numbers from the study, such as the sample size and key findings. 

• What Do These Findings Mean? Authors should reflect on the new knowledge generated by the research and the implications for practice, research, policy, or public health. Authors should also consider how the interpretation of the study’s findings may be affected by the study limitations. In the final bullet point of ‘What Do These Findings Mean?’, please describe the main limitations of the study in non-technical language.

Please amend as outlined above.

INTRODUCTION

Please conclude the Introduction with a clear description of the study question or hypothesis.

METHODS and RESULTS

Please report the number of [patients, samples, etc] and dates of recruitment, and account for all methods used in your study.

Please define "lost to follow-up" as used in this study. Other reasons for exclusion should be defined.

Please define the length of follow up (eg, in mean, SD, and range).

Please provide the actual numbers of events for the outcomes, not just summary statistics or ORs.

Please present numerators and denominators used to derive percentages.

Please ensure to indicate where analyses are adjusted and which factors are adjusted for.

As for the abstract, please ensure to quantify the main results with 95% CIs and p values.

When a p value is given, please specify the statistical test used to determine it.

The methods section is very long and it may help to improve reader accessibility if some of the details could be moved to the supporting information files.

Line 242 – Study cohorts and data preprocessing – this section is particularly long and contains information regarding datasets and statistical methods. Suggest separating this information into two different sections (databases/cohorts and statistical methods) to improve reader accessibility.

Line 529 – suggest ‘Characteristics of the study population’ as an alternative sub-heading.

TABLES and FIGURES

Please see here for guidelines on submitting and citing figures https://journals.plos.org/plosmedicine/s/figures#loc-how-to-submit-figures-and-captions

Please provide titles and legends for all tables and figures (including those in Supporting Information files).

Please ensure that each table and figure is affiliated to a caption which clearly describes the figure content without the need to refer to the text.

Please ensure that all abbreviations are defined in the caption or an appropriate footnote, including those used to report statistical information.

Please ensure to include the meaning of any dots/lines/bars.

Please consider avoiding the use of red and green in order to make figures more accessible to those with colour blindness.

To help facilitate transparent data reporting, where adjusted analyses are presented please also present the unadjusted analyses for comparison. In a caption or footnote please detail all factors adjusted for.

DISCUSSION

Previously we asked, ‘Please present and organize the Discussion as follows: a short, clear summary of the article's findings; what the study adds to existing research and where and why the results may differ from previous research; strengths and limitations of the study; implications and next steps for research, clinical practice, and/or public policy; one-paragraph conclusion.’

Reading through, the discussion is very long and it is easy to get lost. Please revise for brevity and improve accessibility ensuring that the above structure is followed and the items easily identifiable. Pleas avoid the use of subheadings.

REFERENCES

For in-text reference callouts please place citations in square parentheses separate by commas. For example, [1,3,6] or [1-3]. Please check and amend throughout all sub-sections of the manuscript and supporting files.

In the bibliography please ensure that you list up to but no more than 6 author names followed by et al.

For all web references please ensure you include an, ‘Accessed [date].’

Journal name abbreviations should be those listed in the National Center for Biotechnology Information (NCBI) databases.

Line 881 – please remove the data availability statement from the main manuscript and include only in the manuscript submission form when you resubmit the manuscript. It will be compiled as metadata at the time of publication.

SUPPORTING INFORMATION

In the published article, supporting information files are accessed only through a hyperlink attached to the captions. For this reason, you must list captions at the end of your manuscript file. You may include a caption within the supporting information file itself, as long as that caption is also provided in the manuscript file. Do not submit a separate caption file.

As the supporting information files are contained with a single file: 

Please label the file as ‘S1 Supporting Information’.

Please apply alphabetical labelling to each table and figure contained within the S1 file. For example, ‘Fig A’ to ‘Fig Z’ and ‘Table A’ to ‘Table Z’.

Plain text does not need to be labelled and can just be given a title as necessary. For example, ‘Statistical Analysis Plan’.

Please cite tables/figures as ‘Fig A in S1 Supporting Information’ and/or ‘Table A in S1 Supporting Information’, for example.

Please cite plain text as, ‘Statistical Analysis Plan in S1 Supporting Information’, for example.

Alternatively, you may upload each file in the supporting information as individual files and label tables as ‘S1 Table’ (so on) and figures as ‘S1 Fig’ (and so on).

Any additional documents (protocols or appendices) can be labelled as ‘S1 Appendix’, for example.

Please cite items as exactly as labelled.

Further guidance can be found here https://journals.plos.org/plosmedicine/s/supporting-information

SOCIAL MEDIA

To help us extend the reach of your research, please detail any X (formerly Twitter) handles you wish to be included when we tweet this paper (including your own, your coauthors’, your institution, funder, or lab) in the manuscript submission form when you re-submit the manuscript.

Comments from Reviewers:

Reviewer #1: Comments have been addressed 

Perhaps some additional information can be provided regarding the variant rs554533790 (which genes it is close to) - if at all possible. 

Reviewer #2: We thank the authors for largely addressing our previous concerns. On the point where one individual from inferred kinship relationships being retained, it might be detailed as to how the retained individual was selected - was it essentially random (i.e. first encountered individual retained, then any of their relations excluded)?

Reviewer #3: The authors have addressed all of my comments. Thanks!

[LINK]

---

## [Editor Report · Decision Letter 3]

23 Jul 2024

Dear Dr Liu, 

On behalf of my colleagues and the Academic Editor, Professor Christelle Nguyen, I am pleased to inform you that we have agreed to publish your manuscript "Variability in performance of genetic-enhanced DXA-BMD prediction models trained on the UK Biobank across ethnic and geographical populations: A genetic risk prediction study" (PMEDICINE-D-24-00964R3) in PLOS Medicine.

Prior to publication, when completing the required formatting changes (detailed below), please make the following amendments:

1) Title – please revise the title to read as follows: “Variability in performance of genetic-enhanced DXA-BMD prediction models across diverse ethnic and geographic populations: A risk prediction study”.

2) Author summary – line 85 – please define ‘BMD’ and ‘(DXA)’.

3) Tables – please include the full names of the cohorts either in column 1 or defined in the footnote below.

PRESS

Thank you again for submitting to PLOS Medicine, it has been a pleasure handling your manuscript. We look forward to publishing your paper. 

Kind regards,

Pippa 

Philippa Dodd, MBBS MRCP PhD 

Senior Editor 

PLOS Medicine

pdodd@plos.org